# Multi-model study of HTAP II on sulphur and nitrogen deposition

Jiani Tan[1], Joshua S. Fu[1], Frank Dentener[2], Jian Sun[1], Louisa Emmons[3], Simone Tilmes[3], Kengo Sudo[4], Johannes Flemming[5], Jan Eiof Jonson[6], Sylvie Gravel[7], Huisheng Bian[8], Yanko Davila[9], Daven K. Henze[9], Marianne T Lund[10], Tom Kucsera[11], Toshihiko Takemura[12], Terry Keating[13]

[1] Department of Civil and Environmental Engineering, University of Tennessee, Knoxville, TN, USA
[2] European Commission, Institute for Environment and Sustainability Joint Research Centre, Ispra, Italy
[3] Atmospheric Chemistry Observations and Modeling Laboratory, National Center for Atmospheric Research, Boulder, Colorado, USA
[4] Nagoya University, Furo-cho, Chikusa-ku, Nagoya, Japan
[5] European Centre for Medium-Range Weather Forecasts, Reading, UK
[6] Norwegian Meteorological Institute, Oslo, Norway
[7] Meteorological Research Branch, Meteorological Service of Canada, Toronto, Canada
[8] National Aeronautics and Space Administration Goddard Space Flight Center, Greenbelt, MD, USA
[9] Department of Mechanical Engineering, University of Colorado, Boulder, CO, USA
[10] CICERO Center for International Climate Research, Oslo, Norway
[11] Universities Space Research Association, NASA/GESTAR, Columbia, MD, USA
[12] Research Institute for Applied Mechanics, Kyushu University, Fukuoka, Japan
[13] US Environmental Protection Agency, Washington, DC, USA

*Correspondence to:* Joshua S. Fu (jsfu@utk.edu)

**Abstract.** This study uses multi-model ensemble results of 11 models from the 2$^{nd}$ phase of Task Force Hemispheric Transport of Air Pollution (HTAP II) to calculate the global sulfur (S) and nitrogen (N) deposition in 2010. Modelled wet deposition is evaluated with observation networks in North America, Europe and Asia. The modelled results agree well with observations, with 76-83% of stations having predicted within ±50% of observations. The results underestimate $SO_4^{2-}$, $NO_3^-$ and $NH_4^+$ wet depositions in some European and East Asian stations, but overestimate $NO_3^-$ wet deposition in Eastern United States. Inter-comparison with previous projects (PhotoComp, ACCMIP and HTAP I) shows HTPA II has considerably improved the estimation of deposition at European and East Asian stations. Modelled dry deposition is generally higher than the "inferential" data calculated by observed concentration and modelled velocity in North America, but the inferential data has high uncertainty, too. The global S deposition is 84 Tg(S) in 2010, with 49% of the deposits on continental regions and 51% on ocean (19% on coastal). The

global N deposition consists of 59 Tg(N) oxidized nitrogen ($NO_y$) deposition and 64 Tg(N)
reduced nitrogen ($NH_x$) deposition in 2010. 65% of N is deposited on the continental regions and
35% is on ocean (15% on coastal). The estimated outflow of pollution from land to ocean is
about 4 Tg(S) for S deposition and 18 Tg(N) for N deposition. Compared our results to the
results in 2001 from HTAP I, we find that the global distributions of S and N depositions have
changed considerably during the last 10 years. The global S deposition decreases 2 Tg(S) (3%)
from 2001 to 2010, with significant decreases in Europe (5 Tg(S) and 55%), North America (3
Tg(S) and 29%) and Russia (2 Tg(S) and 26%), and increases in South Asia (2 Tg(S) and 42%)
and the Middle East (1 Tg(S) and 44%). The global N deposition increases by 7 Tg(N) (6%),
mainly contributed by South Asia (5 Tg(N) and 39%), East Asia (4 Tg(N) and 21%) and
Southeast Asia (2 Tg(N) and 21%). The $NH_x$ deposition is increased with no control policy on
$NH_3$ emission in North America. On the other hand, $NO_y$ deposition starts to dominate in East
Asia (especially China) due to boosted $NO_x$ emission.
**1 Introduction**
The nitrogen (N) plays an important role in the balance of the global ecosystem. Human
activities such as consumption of fossil fuels, production and usage of N fertilizers and livestock
cultivation disturb the N cycle in the ecosystem (Vitousek et al., 1997; Galloway et al., 2008).
Estimation under the IPCC SRES A2 scenario predicts that the N deposition over land will
increase by a factor of ~2.5 from 2000 to 2100 (Lamarque et al., 2005). Elevated N deposition
can cause exceedance of N critical loads on ecosystems (Sanderson et al., 2006; Sun et al., 2017).
11% of the world's natural vegetation has already received N deposition that exceeds the critical
load in 2000 (Dentener et al., 2006). The most affected regions are Eastern Europe (80%), South
Asia (60%) and East Asia (40-50%). This percentage will be 40% for the world's protected areas
in 2030 (Bleeker et al., 2011). Elevated S and N deposition are also associated with a host of
environmental issues such as acidification and eutrophication of the terrestrial system (Bouwman
et al., 2002), loss of ecosystem biodiversity (Bobbink et al., 2010), harming the heterotrophic
respiration and disturbing the soil decomposition process (Janssens et al., 2010), although some
studies found increasing N deposition could benefit the carbon uptake by land processes (Reay
et al., 2008; Holland et al., 1997). Similar to the terrestrial system, over-richness of S and N
deposition is also a threat to the aquatic system by acidification (Doney et al., 2007) and
eutrophication of the ocean (Bergstrom and Jansson, 2006; Jickells, 2006; Jickells et al., 2017).
In order to understand S and N deposition, a number of global scale studies have been
conducted in the last decade. Dentener et al. (2006) investigated the current (2000) and future
(2030) S and N deposition with multi-model ensemble results of ACCENT IPCC-AR4
experiment (PhotoComp). Model evaluation showed that 60-70% of modelled wet deposition is
within ±50% of measurements in Europe and North America. $NH_x$ deposition was overestimated
in South Asia and $NO_y$ deposition was underestimated in East Asia. 11% of the world's nature
vegetation received N deposition that exceed the critical load in 2000, and this percentage would
increase to 17% under current air quality legislation and 25% under IPCC SRES A2 scenario in
2030. Sanderson et al. (2008) used the ensemble results of the 1st phase of the Task Force
Hemispheric Transport of Air Pollution (HTAP I) to estimate the long-range transport of
oxidized nitrogen between Europe, North America, South Asia and East Asia. Results showed
that 8-15% of $NO_x$ from source regions could be transported beyond the distance of 1000 km,
which indicated the impact of inter-continental transport of air pollutants on deposition.
Lamarque et al. (2013) calculated the S and N deposition in 2000 using a multi-model ensemble
of the Atmospheric Chemistry and Climate Model Intercomparison Project (ACCMIP). Model
performance on $NO_3^-$ wet deposition was comparable with PhotoComp and HTAP I, but $NH_4^+$
wet deposition was not well simulated. Simulations with the projected emissions in 2100 under
four Representative Concentration Pathways (RCP) indicated that N deposition is likely to
substantially increase in Latin America, Africa and parts of Asia (especially South Asia) in the
future. Vet et al. (2014) conducted a comprehensive evaluation on the model performance of
HTAP I. The results underestimated the wet deposition at observation sites with high observed N
deposition in North America, Southern and Northern Europe and East Asia. Dry deposition in the
Unites States was found to deviate with inferential dry deposition data. Kanakidou et al. (2016)
used the ACCMIP simulation results under historical, RCP6.0 and RCP 8.5 emission scenarios to
estimate the changes in N deposition driven by human activity in the past (1850), present (2005)
and future (2050). Their results showed that organic nitrogen (ON) from primary emission and
secondary organic aerosol (SOA) account for 20-30% of total N deposition. The impact of
human activity on N deposition has increased from 15% in the past to 60% in present years.
This impact was likely to persist in the future. Bian et al. (2017) examined the possible factors
causing the inter-model diversity in simulating $NO_3^-$ and $NH_4^+$ deposition by comparing the
results of 9 models participating in the $3^{rd}$ phase of Aerosol Comparisons between Observations
and Models (AeroCom III). The results showed that models have large differences in calculating
the pH adjustment for the effective Henry's law constant, which could largely influence the
simulation of $NH_x$ wet deposition.
These studies give a clear view to S and N deposition in the early 2000s. However, large
changes are seen in the global N emissions in the last decade (van der A et al., 2008), including a
large increase in China (Zhang et al., 2009b; van der A et al., 2006; Richter et al., 2005;
Kurokawa et al., 2013; Zhang et al., 2007; Li et al., 2017), and general decreases in both Europe
(Tørseth et al., 2012) and Eastern United States (Kim et al., 2006). In addition, ground
observations and satellite measurements show large increases in the dry deposition in the western
United States, Eastern Europe and East China, together with decreases in Eastern United States,
Western Europe and Japan (Jia et al., 2016). Thus, a follow-up study is needed to update our
knowledge about the S and N deposition with emission changes in the $21^{st}$ century.
In this study, we use the multi-model mean (MMM) of 11 global models from the $2^{nd}$
phase of HTAP (HTAP II) project to calculate the S and N deposition in 2010. Section 2 gives a
short description of the HTAP II project and introduces the method to develop MMM and
metrics for model evaluation. Section 3.1 evaluates MMM performance on wet deposition with
observations from networks in North America, Europe and East Asia. The modelled dry
deposition is compared with the inferential data in North America (see detail in Section 3.1). We
also compare the model performance of this study with previous studies in 2001 of PhotoComp
(Dentener et al., 2006), HTAP I (Vet et al., 2014), and ACCMIP (Lamarque et al., 2013). Section
3.2 and Section 3.3 estimate the S and N deposition on continental, coastal and ocean regions in
2010. By comparing our results with deposition in 2001 of HTAP I, we investigate the changes
of deposition in the past 10 years. We conclude with the findings in Section 4.
**2   Methodology**
**2.1  Model description and Experiment setup**
The HTAP was developed in 2005 aiming at understanding the long-range transport of air
pollution and its impact on regional air quality. HTAP I has involved more than 20 global
models with base simulation year of 2001. A comprehensive assessment has been published to
summarize the findings in HTAP I with respect to the long-range transport of (1) ozone and
particulate matter (2) mercury and (3) persistent organic pollutants (HTAP, 2010). The HTAP II
was launched in 2012 with base year of 2010. A prescribed emission inventory called HTAPv2.2
is used by models from different groups to facilitate a fair evaluation of the models' ability and
uncertainty (Galmarini et al., 2017). It is a harmonized emission inventory formed by the best
estimation of emissions from different organizations, including Environmental Protection
Agency (EPA) of United States, the EPA and Environment Canada, the European Monitoring
and Evaluation Programme (EMEP) and the Netherlands Organisation for Applied Scientific
Research (TNO), the Model Inter-Comparison Study for Asia (MICS-Asia III) and the Emission
Database for Global Atmospheric Research (EDGARv4.3). The development of the emission
inventory is described in Janssens-Maenhout et al. (2015). Following are some highlight findings
in HTAP II. Stjern et al. (2016) estimated the impact of domestic and foreign emission change of
BC, OC and $SO_4$ on regional radiative forcing. Huang et al. (2017) studied the impact of
intercontinental outflow from East Asia to North America on $O_3$ pollution by simulating the
regional-scale Sulfur Transport and dEposition Model (STEM) with boundary conditions
provided by 3 global transport models. Jonson et al. (2018) conducted a source apportionment
for $O_3$ pollution in Europe and calculated the contributions of emission from global wide. Tan et
al. (2018) investigated the intercontinental export of sulfur and nitrogen emission and its impact
on local deposition.
Among the 20 models participating in the HTAP II project (configurations described in
Stjern et al. (2016)), 11 models (i.e. CAM-Chem, CHASER_re1, CHASER_t106, EMEP_rv48,
GEMMACH, GEOS5, GEOSCHEMAJOINT, OsloCTM3v.2, GOCARTv5, SPRINTARS and
C-IFS_v2) submitted the model outputs of S and N deposition. To develop the MMM, all models
are interpolated to a uniform 0.1º × 0.1° horizontal resolution (the same resolution as the
emission inventory) by linear interpolation. Then the MMM of the emission/deposition quantities
of each of S and N is calculated by averaging (arithmetic mean) all available model outputs.
More details are demonstrated in Section 2.2. The base year of simulation is 2010, with
additional six-month run as model spin-up. The administrative boundaries of 17 regions are
shown in Fig. S1. Details about the experiment setup can be found in Galmarini et al. (2017).

## 2.2 Method for calculating the MMM

To make the discussion clear, we define the terms as follows: The continental regions refer to all land regions including the Antarctic. The coastal regions are defined in Fig. S1. In section 3.2 and 3.3, the S deposition contains gas phase $SO_2$ deposition and aerosol $SO_4^{2-}$ deposition. The N deposition includes oxidized nitrogen ($NO_y$) deposition and reduced nitrogen ($NH_x$) deposition. $NO_y$ deposition is composed of all oxidized nitrogen species except $N_2O$. Based on the model outputs, $NO_y$ deposition mainly includes $NO_2$, $HNO_3$, aerosol $NO_3^-$, peroxyacyl nitrate (PAN) and other organic nitrates than PAN (Orgn). $NH_x$ deposition consists of gas phase $NH_3$ deposition and aerosol $NH_4^+$ deposition. Before constructing the MMM, we check the quality of model outputs using two criteria. First, we check the mass balance of each of the models by comparing the global deposition of each with its emission. Models are excluded if their deposition values fall outside the range of ±20% of their emission values. The second criterion is to check if the result of a model is away from the mean value of all models. We adopt the upper and lower limits as median of models ± 1.5 × interquartile by Vet et al. (2014) and check the values separately for all species of deposition and emission. The models used to develop the MMM and their values are summarized in Table S1-S3. After the quality check, we calculate the mean value of each species using equation (1) with all available model outputs. Then, we combine all of the related species into total deposition/emission by equation (2).

$$S_{MMM}(j) = \frac{1}{n}\sum_{i=1}^{n} S_i(j) \tag{1}$$

$$S_{MMM}(NO_y, NH_x \text{ or } S) = \sum_{j=1}^{s} S_{MMM}(j) \tag{2}$$

For both equations (1) and (2), $i$ is the individual model and $j$ is the species of deposition/emission from model outputs. $S_i(j)$ is the species $j$ from model $i$ and $S_{MMM}(j)$ is the MMM of species $j$.

## 2.3 Model evaluation metrics

To compare the model performance with previous projects consistently, we adopt the following metrics in Lamarque et al., (2013): Linear fit slope, mean bias, mean observation, mean model, correlation coefficient (R) and fraction (of model results) within ± 50% (of observations).

In addition, we use 4 statistical metrics following Eq. (3)-(6).

$$\text{NMB (normalized mean bias)} = \frac{\sum_{i=1}^{n}(M_i - O_i)}{\sum_{i=1}^{n} O_i} \times 100 \tag{3}$$

$$\text{NME (normalized mean error)} = \frac{\sum_{i=1}^{n}|M_i - O_i|}{\sum_{i=1}^{n}O_i} \times 100 \qquad (4)$$
$$\text{MFB (mean fractional bias)} = \frac{1}{n}\sum_{i=1}^{n}\frac{M_i - O_i}{(M_i + O_i)/2} \times 100 \qquad (5)$$
$$\text{MFE (mean fractional gross error)} = \frac{1}{n}\sum_{i=1}^{n}\frac{|M_i - O_i|}{(M_i + O_i)/2} \times 100 \qquad (6)$$
For equations (3)-(6), $M_i$ is the model result, $O_i$ is the observation and n is the sample size.
NMB, NME, MFB and MFE normalize the model mean bias to avoid data inflation in case of
large data range. NMB and NME normalize the mean bias by the observation data and thus may
tend toward model overestimation. MFB and MFE normalize the mean bias by the average of
observations and model results, considering both model overestimation and underestimation, and
thus are less biased. In Section 3.1, we use MFB and MFE as the main metrics to evaluate the
model performance.
**3   Results**
**3.1  Evaluation of model performance**
**3.1.1   Wet deposition**
We evaluate the MMM results of $SO_4^{2-}$, $NO_3^-$ and $NH_4^+$ wet deposition with site observations in
United States, Europe and East Asia. The MMM result is annual deposition in 2010 and the
observation data is 3-year annual average deposition during 2009-2011. The observation data in
United States comes from the National Atmospheric Deposition Program (NADP)
(http://nadp.sws.uiuc.edu/). The quality and completeness of the observations are checked
according to the 4 criteria established by the NADP technical committee
(http://nadp.sws.uiuc.edu/documentation/notes-depo.html). As a result, we use the data from 136
stations of the 267 available stations. The observations in Europe are derived from the European
Monitoring and Evaluation Programme (EMEP) CCC reports
(http://www.nilu.no/projects/ccc/reports.html). After checking the data quality and completeness,
we use the data from 82 stations of the 102 available stations. The observations in Asia are from
the Acid Deposition Monitoring Network in East Asia (EANET) (http://www.eanet.asia/). Data
from 43 stations of the 52 available stations are used for evaluation.

Fig. 1 shows the scatter plots of the MMM $SO_4^{2-}$, $NO_3^-$ and $NH_4^+$ wet deposition with

observations at the NADP, EMEP and EANET stations. Performances of individual models can
be found in Fig. S2-S4 in the supplementary material. The $SO_4^{2-}$ wet deposition comprises gas
phase $SO_2$ and aerosol $SO_4^{2-}$ wet deposition. The $NO_3^-$ wet deposition includes gas phase $HNO_3$
and aerosol $NO_3^-$ wet deposition. The $NH_4^+$ wet deposition contains gas phase $NH_3$ and aerosol
$NH_4^+$ wet deposition. Performance of individual models can be found in Figs. S2-S4 in the
supplementary material. Fig. 2 displays the spatial distributions of MMM $SO_4^{2-}$, $NO_3^-$ and $NH_4^+$
wet deposition (contours) with observations (filled circles). In terms of $SO_4^{2-}$ wet deposition, the
MMM results are consistent with observations at the NADP stations with a close to 1 slope (0.9)
and a high R value (0.8) (Fig.1 (a)). The MFB and MFE are 9% and 32%, indicating slight
overestimation. According to Fig. 2(a), the observed $SO_4^{2-}$ wet deposition is highest in
northeastern United States, and this spatial distribution is well captured by MMM. The EMEP
stations are well simulated with low MFB (-7%) and MFE (25%) (Fig. 1(b)). The MMM
predictions are within ±50% of observations at 87% of the stations. According to Fig. 2(b), 1
station in Poland and 1 station in Norway, with observed $SO_4^{2-}$ wet deposition of 1000 and 500
mg (S) $m^{-2}$ $yr^{-1}$ respectively, are underestimated by 50%. We evaluate the model performances
on simulating precipitation (Fig. S5 and Fig. S6). For the Norway site, the observed precipitation
is 1566 mm $yr^{-1}$ and the mmm underestimated the precipitation by 49%, which fits well for the
50% underestimation of $SO_4^{2-}$ wet deposition at this site. For the Polish site, the observed
precipitation is 1137 mm $yr^{-1}$ and the mmm underestimated the precipitation by 21%. The
underestimation in precipitation could partly explain the negative model bias in simulating $SO_4^{2-}$
wet deposition. Another possible reason is the high topography of the sites. The Polish site is
1603 meters above sea, which is one of the highest sites among the European sites. Similar to the
Polish site, one site in Spain, which is 1360 meters height, is underestimated by 142 mg (S) $m^{-2}$
$yr^{-1}$ (59%) for $SO_4^{2-}$ wet deposition, while its precipitation is well simulated with a slight positive
model bias of 5%. At the EANET stations, very high $SO_4^{2-}$ concentrations were measured at
some stations, probably correlated with dust emission (Dentener et al., 2006). Therefore, we
ignore the measurements coincident with measured calcium ($Ca^{2+}$) deposition larger than 20
mole $m^{-2}$ $yr^{-1}$. The evaluation (Fig. 1(c)) shows that the $SO_4^{2-}$ wet deposition is generally
underestimated at the EANET stations by 23% (MFB) and 44% (MFE). The stations in Korea
and Vietnam are generally underestimated by more than 200 mg (S) $m^{-2}$ $yr^{-1}$ (Fig. 2(c)). On the
other hand, the $SO_4^{2-}$ wet deposition is generally well simulated in Indonesia, Philippines,
Thailand and Japan. Overall, 76% of the stations of all networks predicted quantities within
±50% of observations. The EANET stations have the highest model bias among the 3 networks.
It should be noted that for the 3 excluded stations (located in China) with high $Ca^{2+}$ deposition,
the $SO_4^{2-}$ wet deposition is largely underestimated by more than 1000 mg (S) $m^{-2}$ $yr^{-1}$ (not shown
in figures). If we include these stations in the model evaluation, the mean bias for East Asia
changes from -160 mg (S) $m^{-2}$ $yr^{-1}$ to -300 mg (S) $m^{-2}$ $yr^{-1}$. We also realize that the observation
stations in China are mainly located along the eastern and southern coast, while the highest
modelled deposition is found in the inland areas. Therefore, it is hard to conduct a
comprehensive evaluation over this region due to unavailable measured data in the inland areas.

For $NO_3^-$ wet deposition, the MMM results agree well with observations at the NADP

stations, as shown by the linear regression line in Fig. 1(d) with slope of 1.2 and R value of 0.9.
However, the amount of deposition is overestimated by 33% (MFB) and 36% (MFE). According
to Fig. 2(d), there is a general tendency of overestimation throughout the stations in United
States, especially the stations located in Midwest and Southeast. At the EMEP stations, the $NO_3^-$
wet deposition is well simulated with low MFB of -5% and MFE of 24% (Fig. 1(e)). The
modelled deposition is within ±50% of observed deposition at more than 90% of the stations.
The MMM results are close to the observations at stations with deposition lower than 400 mg
(N) $m^{-2}$ $yr^{-1}$, but generally underestimate the deposition at stations with higher observations.
According to Fig. 2(e), wet deposition at 3 stations in Poland, Norway and Spain were
underestimated by 430 (59%), 420 (63%) and 290 (67%) mg N $m^{-2}$ $yr^{-1}$, respectively. Besides,
the stations in Germany generally under-predict these values by 100-200 mg (N) $m^{-2}$ $yr^{-1}$. The
$NO_3^-$ wet deposition at the EANET stations is well simulated with MFB (-3%) and MFE (43%)
(Fig. 1(f)). The model estimations are within ±50% of observations for 77% of the stations.
According to Fig. 2(f), 1 station in Central China is overestimated by 400 (130%) mg (N) $m^{-2}$ $yr^{-}$
$^1$. On the contrary, 3 stations in Thailand, Vietnam and Malaysia are underestimated by 570
(78%), 350 (66%) and 200 (64%) mg (N) $m^{-2}$ $yr^{-1}$. Overall, 83% of the MMM results are within
±50% of observations at stations of all networks. The NADP stations have the highest MFB due
to a generally positive bias in the eastern United States. The EANET stations have the highest
MFE value, mainly due to the underestimation in Southeast Asia.

The modelled $NH_4^+$ wet deposition agrees well with observations at the NADP stations

with MFB of 7% and MFE of 25% (Fig. 1(g)). 88% of modelled deposition is within ±50% of
observations as shown by the R value of 0.9. The MMM has well captured the high deposition in
the United States Midwest, but slightly underestimates the deposition in the Southeast (Fig.
2(g)). At the EMEP stations, the $NH_4^+$ wet deposition is well simulated with MFB of -1% and
MFE of 36% (Fig. 1(h)). The MMM results are close to the observations at most stations and
well reproduce the high deposition in Germany and Italy (Fig. 2(h)). Some stations in Norway
and Poland are slightly underestimated by 100-200 mg (N) $m^{-2}$ $yr^{-1}$. These stations all report
observations of higher deposition than 500 mg (N) $m^{-2}$ $yr^{-1}$. The $NH_4^+$ wet deposition is
underestimated at the EANET stations by 10% (MFB) and 50% (MFE) (Fig. 1(i)). The MMM
has well captured the high deposition in Eastern China and Indonesia, but generally
underestimates the $NH_4^+$ wet deposition at the Russian stations (Fig. 2(i)). In addition, the
observed deposition at the 3 Korean stations is relatively high (~500-600 mg (N) $m^{-2}$ $yr^{-1}$), but
the MMM fails to reproduce any of them. There could be a missing emission source in that
region. Overall, 81% of the MMM predictions are within ±50% of observations at stations of all
networks. The $NH_4^+$ wet deposition is somewhat underestimated in all 3 regions, especially in
East Asia.

Table 1 compares the model performance of this study (HTAP II) with previous projects

of PhotoComp (Dentener et al., 2006), HTAP I (Vet et al., 2014) and ACCMIP (Lamarque et al.,
2013). It should be noted that the emission inputs, simulation periods and participating groups of
this study (year 2010) are different from those of the previous projects (year 2001). Although the
observations are from the same networks, the previous projects used 3-year averaged
observations of 2000-2002 and this study used those of 2009-2011. Due to these differences, the
model performances may not be totally comparable. In terms of $SO_4^{2-}$ wet deposition, the model
performance is similar to that for previous projects in North America, with 4-6% higher
percentage of stations within ±50% of observations. Large improvement is found in Europe. The
absolute mean bias decreases from 50-130 mg (S) $m^{-2}$ $yr^{-1}$ to 30 mg (S) $m^{-2}$ $yr^{-1}$. There is 10%
increase in the fraction of stations within ±50% of observations. At the East Asian stations, the
absolute mean bias decreases slightly from 180~290 mg (S) $m^{-2}$ $yr^{-1}$ to 160 mg (S) $m^{-2}$ $yr^{-1}$. But
the R value and fraction within ±50% have somewhat declined.  For $NO_3^-$ wet deposition, HTAP
II performs similar to the ensembles used in previous projects in North America, but slightly
better in Europe with lower mean bias and 5% increase in the fraction within ±50% of
observations. The model mean bias at the Asian stations has decreased significantly from ~50
mg (N) $m^{-2}$ $yr^{-1}$ to ~1 mg (N) $m^{-2}$ $yr^{-1}$. However, the biases for individual models are large (Fig.
S3). Large negative model bias is found in Southeast Asia and improvements are needed in the
future. In terms of $NH_4^+$ wet deposition, HTAP II shows similar R values to those of ensembles
used for the previous projects at the NADP stations, with slightly lower model bias. However,
HTAP II shows considerable improvement in Europe. The slope of the regression line increases
from 0.3-0.4 to 0.6 and the mean bias decreases from as large as -95 mg (N) $m^{-2}$ $yr^{-1}$ to -4 mg (N)
$m^{-2}$ $yr^{-1}$. For Asia, the slope, mean bias and R values for HTAP II are all within the ranges of the
previous projects, while the absolute mean bias decreases form 70~140 mg (N) $m^{-2}$ $yr^{-1}$ to 30 mg
(N) $m^{-2}$ $yr^{-1}$.
**3.1.2 Dry deposition**
The number of dry deposition measurements is limited due to difficulty in measuring the dry
deposition directly by instruments. This study evaluates the dry deposition in United States using
information from the Clean Air Status and Trends Network (CASTNET). Instead of direct
measurements, the data are produced by an "inferential" method, using calculations of the
measured concentration of species and modelled dry deposition velocities. We use the 3-year
average data of 2009-2011 from CASTNET and adopt the same selection criteria as we did for
the wet deposition measurements. Data from 81 stations out of 85 available stations is used for
comparison. Fig. 3 shows the scatter plots of the MMM $SO_2$, $SO_4^{2-}$, $NO_3^-$, $HNO_3$ and $NH_4^+$ dry
deposition with inferential data at the CASTNET stations. Performances of individual models
can be found in Fig. S7-S11 in the supplementary material. The modelled $SO_2$ dry deposition is
240 (170%) mg (S) $m^{-2}$ $yr^{-1}$ higher than the inferential data and only 5% of the stations is within
±50% of the inferential values. There are smaller discrepancies for values of $SO_4^{2-}$ dry deposition
(14 mg (S) $m^{-2}$ $yr^{-1}$ and 60%) between model and inferential results. Modelled $NO_3^-$, $HNO_3$ and
$NH_4^+$ dry deposition is generally 50-100% higher than the inferential data and the fraction within
±50% is about 15%. Fig. 4 shows the spatial distributions of MMM dry deposition (contours)
with the inferential data (filled circles). The MMM results are consistent with the inferential data
in the western United States, where the dry deposition is generally low. And both datasets predict
high $NO_3^-$ dry deposition in western California. Large disagreements are found in the eastern
United States. In the Midwest (mainly Indiana and Ohio states), although both results estimate
higher N ($NO_3^-$, $HNO_3$ and $NH_4^+$) dry deposition in this region than the others, the prediction of
MMM is 20-30 mg (N) $m^{-2}$ $yr^{-1}$ higher than the inferential data at every station. In addition, the
MMM estimates much higher deposition in southern and northeastern United States than in the
western United States, but this gradient is much weaker in the inferential data.
Table 2 compares the model performance of this study (HTAP II) with that of the models
used in the previous projects of HTAP I (Vet et al., 2014) and ACCMIP (Sun et al., 2017).
HTAP I used the 2001 simulation results and compared them with 3-year average (2000-2002)
CASTNET data. ACCMIP used 10-yr averages of both MMM and CASTNET data from 2000 to
2009. The N dry deposition values for all projects contain $NO_3^-$, $NH_4^+$ and $HNO_3$ and the S dry
deposition includes $SO_2$ and $SO_4^{2-}$. Both HTAP I and HTAP II overestimated the S and N dry
deposition, but HTAP II has ~100 mg(S) m$^{-2}$ yr$^{-1}$ and ~80 mg(N) m$^{-2}$ yr$^{-1}$ lower mean bias than
HTAP I. The comparison with ACCMIP results may not be solid since there are large differences
in simulation periods. Generally, the HTAP II performance is similar to ACCMIP for $NH_4^+$, $SO_2$
and $SO_4^{2-}$ dry deposition simulation, but has larger mean bias for $HNO_3$ dry deposition
prediction.
Since the CASTNET dry deposition is not actually measured but instead a calculation of
measured concentration of species and modelled dry deposition velocities, it is necessary to
investigate which factor of these two contributes to the model bias. We compare the modelled air
pollutant concentrations with CASENET measurements as shown in Table S4-S8. The MMM
overestimates the $SO_2$, $SO_4^{2-}$, $HNO_3$, $NO_3^-$ and $NH_4^+$ concentrations by 394%, 40%, 217%,
135% and 173%, respectively. It should be noted that the CASTNET sites are generally located
in rural regions that are away from emission sources (Sickles and Shadwick, 2008), thus the
measured concentrations of air pollutants are relatively low compared with those of urban sites.
While the resolutions of the HTAP II models range from 0.5° to 3°, and are not fine enough to
reproduce the characteristic of some rural sites. The models with finer resolutions except
CHASER_t106 model (i.e. EMEP_rv48 (0.5° × 0.5°) and SPRINTARS (1.1° × 1.1°)) generally
perform better than the others, while models with coarse resolutions (i.e. CHASER_re1 (2.8° ×
2.8°) and OsloCTM3.v2 (2.8° × 2.8°)) are generally not performing well for all species. This
could explain the overestimation of air pollutant concentrations at the CASTNET sites.
In order to check the differences of modelled dry deposition velocity between CASNET
and HTAP II models, we adopt the general approach for calculating dry deposition velocity from
Wesely, (1989).
$$V_d = -F_c / C_a \qquad (7)$$
Where $V_d$ is the deposition velocity, $F_c$ is the dry deposition flux and $C_a$ is the concentration of
species. The negative mark indicates the direction of the dry deposition velocity. This scheme
has been widely adopted in global models (Wesely and Hicks, 2000) with modifications. We
compare the calculated dry deposition velocity of models and CASTNET (Table S9-S13). The
mean bias of dry deposition velocities for MMM are -8%, 0.3%, 7%, 19% and 2% for $SO_2$, $SO_4^{2-}$
, $HNO_3$, $NO_3^-$ and $NH_4^+$, respectively, which are much lower than those of air pollutants. The
model bias for dry deposition at the CASTNET sites mainly comes from the model over
prediction of air pollutant concentration.

In addition, the CASENET estimation of dry deposition has been reported with

uncertainties. Zhang et al. (2009a) estimated a 10-20% uncertainty in the measurement of mixing
ratio of species, 20% in the calculated velocity and ~20% when lacking of hourly concentration
for species with strong diurnal variation. Schwede et al. (2011) compared CASTNET dry
deposition estimates with those of the Canadian Air and Precipitation Monitoring Network
(CAPMoN). The CASTNET data is 54% lower for $SO_2$ dry deposition and 47% lower for $HNO_3$
dry deposition than CAPMoN, mainly due to using different models to calculate the dry velocity.
**3.2  Total S deposition**

Table 3 lists the calculated amount of S emission and deposition on continents, coastal

regions and oceans. Fig. 5 presents the distribution of S emission and deposition from MMM
results. The distributions of components of S deposition are shown in Fig. S12 in the
supplementary material. The global S deposition is 84 Tg(S) in 2010, with 49% deposits on non-
coastal continents, 32% deposits on non-coastal ocean and 19% deposits on coastal area. For
continental non-coastal regions, East Asia receives the largest amount of S deposition (17%).
The highest S deposition is found in Eastern China (2000 mg(S) $m^{-2}$ $yr^{-1}$) (Fig. 5(b)). Other
regions with largely extended areas of high S deposition are the Indian peninsula (800-1200
mg(S) $m^{-2}$ $yr^{-1}$), Malaysia and Indonesia (~1200 mg(S) $m^{-2}$ $yr^{-1}$), United States Midwest (800-
2000 mg(S) $m^{-2}$ $yr^{-1}$), Mexico and Central America (400-800 mg(S) $m^{-2}$ $yr^{-1}$), Peru and Chile
(400-600 mg(S) $m^{-2}$ $yr^{-1}$), Eastern Europe (~800 mg(S) $m^{-2}$ $yr^{-1}$) and the northeastern Middle
East (500-1200 mg(S) $m^{-2}$ $yr^{-1}$). The distribution of high deposition regions agrees very well with
high S emission regions (Fig. 5(a)). For coastal regions, East Asia and Southeast Asia receive the
most S deposition (3% and 3% respectively). The east coast of East Asia and North America and
all of the coast of India have relatively high deposition (400-800 mg(S) m$^{-2}$ yr$^{-1}$), followed by the
west coast of Mexico (~400 mg(S) m$^{-2}$ yr$^{-1}$). This study estimates 43 Tg(S) of S deposition on the
ocean and coastal regions in 2010, and accounts for 51% of global S deposition. The ratio is
similar to the 51% estimated by Dentener et al. (2006) and 46% estimated by Vet et al. (2014) in

2001.

We calculate the ratio of S deposition to S emission (Fig. 5(c)). Because it is not clear
how dimethyl sulphide (DMS) emission will transfer to S deposition, this ratio does not represent
the transformation of S emission to deposition. For continental non-coastal regions, the average
ratio is 85% (86% if taking consideration of coastal regions). In high emission regions, this ratio
can be viewed as the "scavenging" effect of S pollution by deposition. In major source regions of
S emission (i.e. North China Plain, Midwest of United States and India), the ratios are only
slightly higher than 50%, while in low S emission regions (<10 mg(S) m$^{-2}$ yr$^{-1}$), the ratios could
exceed 400 % (areas with white color in Fig. 5(c)). This result indicates that the deposition in
these regions is largely affected by long-range transport of pollution from other regions. The
impact of intercontinental transport of air pollutants on deposition can be quantified by the
emission perturbation experiments in HATP II. Results from those experiments will be discussed
in another paper (Tan et al., 2018).
We compare the S emission and deposition in 2010 from HTAP II with that in 2001 from
HTAP I (Vet et al., 2014) (Table 3). We re-calculate the HTAP I results according to the regions
defined in HTAP II (Fig. S1), so the HTAP I results may look different from those in Table 2 of
Vet et al. (2014). Because different models were used for each of the two ensembles compared,
associated uncertainty is expected. In addition, emissions in HTAP I were not prescribed, so each
modelling group used its own best estimation of emissions (Sanderson et al., 2008). Conversely,
all models in HTAP II, used the same anthropogenic emission values (although there were still
differences in natural emission). Globally, the S emission decreases by 5 Tg(S) from 2001 to
2010, with 8 Tg(S) (13%) decrease in continental non-coastal regions, 6 Tg(S) (32%) increase in
non-coastal ocean regions and 3 Tg(S) (15%) decrease in coastal regions. For continental non-
coastal regions, there are big drops in S emissions from Europe (6 Tg(S) and 61%), North
America (3 Tg(S) and 34%) and Russia (2 Tg(S) and 44%). On the other hand, South Asia and
Middle East have 2 Tg(S) (56%) and 1 Tg(S) (69%) increase in S emissions. East Asia, one of
the main contributors to S emission seems to show little change between 2001 and 2010.
However, it has experienced large changes during these 10 years, with stable annual increases
from 2000 to 2005 due to increased energy consumption and decreases after 2006 owing to the
successful implementation of the $SO_2$ control policies in China's 11[th] Five-Year-Plan (FYP) (Lu
et al., 2010). For coastal regions, Europe has experienced a 2 Tg(S) (54%) decrease and East
Asia has experienced a 1 Tg(S) (43%) decrease in S emission. Other regions have relatively
small (0-0.6 Tg(S)) changes. The global S deposition decreases by 2 Tg(S), with 5 Tg(S) (11%)
decrease in continental non-coastal regions, 4 Tg(S) (16%) increase in non-coastal ocean regions
and 1 Tg(S) (5%) decrease in coastal regions. The regions with the largest change in deposition
coincide with those having big changes in emission. For instance, Europe experiences 5 Tg(S)
decrease in S deposition with 8 Tg(S) decrease in emission, and South Asia receives 2 Tg(S)
more S deposition with 2 Tg(S) increase in emission. Fig. S13(b) compares the S deposition in
HTAP II with that in HTAP I. Declined S deposition is found in large areas of the eastern United
States and Europe (400-1,500 mg(S) $m^{-2}$ $yr^{-1}$). Regions with increased S deposition are India and
Indonesia (100-800 mg(S) $m^{-2}$ $yr^{-1}$). In China, there is a mixture of both increases and decreases
in S deposition over different areas.  The changes in S depositions agree well with changes in S
emissions (Fig. S13(a)). During China's 11[th] FYP, one of the main technologies to control the
$SO_2$ emission was to install the Flue Gas Desulfurization (FGD) on power plants (Cao et al.,
2009). The effectiveness of this technology in removing $SO_2$ emission varies considerably
regionally, as a result of several factors such as the coverage of FGD technology on power plants,
local reduction targets and stringency of policy implementation by local governments. On the
other hand, new sources of $SO_2$ emission, such as newly built power plants, are found
responsible for the increased S emissions and deposition over some areas in China (Tan et al.,

2017).

**3.3  Total N deposition**

**3.3.1  $NO_y$ deposition**

Table 4 summarizes the $NO_y$ emission and deposition in each region and Fig. 6 presents
the distribution from MMM results. Distributions of components of $NO_y$ deposition are shown in
Fig. S14 in the supplementary material. The global $NO_y$ deposition is 59 Tg(N) in 2010, with
62% of deposits on non-coastal continents, 22% of deposits on non-coastal ocean and 16% of
deposits on coastal areas. For continental non-coastal regions, East Asia receives the largest $NO_y$
deposition (14%). The highest $NO_y$ deposition is found in northeastern China (2000 mg(N) m$^{-2}$
yr$^{-1}$), followed by the Indian peninsula (800-1200 mg(N) m$^{-2}$ yr$^{-1}$), Malaysia and Indonesia (500-
800 mg(N) m$^{-2}$ yr$^{-1}$), Germany, Switzerland and Poland (500-600 mg(N) m$^{-2}$ yr$^{-1}$), northern Sub-
Saharan Africa (300-500 mg(N) m$^{-2}$ yr$^{-1}$), northeastern Middle East (400-500 mg(N) m$^{-2}$ yr$^{-1}$),
United States Midwest (500-600 mg(N) m$^{-2}$ yr$^{-1}$) and Brazil (300-600 mg(N) m$^{-2}$ yr$^{-1}$).

For coastal regions, the east coast of East Asia also receives the largest amount of $NO_y$

deposition (600 mg(N) m$^{-2}$ yr$^{-1}$ and 4%). Relatively high deposition is found on the east coast of
North America (150-400 mg(N) m$^{-2}$ yr$^{-1}$), all of the coast of India (300-500 mg(N) m$^{-2}$ yr$^{-1}$), the
west coast of Europe and all of the coast of Southeast Asia (150-200 mg(N) m$^{-2}$ yr$^{-1}$). This study
estimates 23 Tg(N) of $NO_y$ deposition on the ocean in 2010 (include ocean non-coastal and
coastal), similar to Dentener et al. (2006)'s estimation of 23 Tg(N), Duce et al. (2008)'s
estimation of 14-32 Tg(N) and Vet et al. (2014)'s estimation of 20 Tg(N). About 38% of global
$NO_y$ deposits on the ocean, lower than 43% in PhotoComp (Dentener et al., 2006) and 42% in
HTAP I (Vet et al., 2014), but higher than 30% estimated by Lamarque et al. (2005). It should be
noted that these values partly depend on the land-ocean mask, which may differ among different
studies. For non-coastal ocean regions, the $NO_y$ deposition is 13 Tg(N), accounts for 22% of the
global deposition. While the emission from oceans is only 2 Tg(N), about 4% of global emission.
The difference of 11 Tg(N) indicates $NO_y$ transport from continents to the open ocean. Antarctic
have near zero $NO_x$ emission, but receive 0.1 Tg(N) $NO_y$ deposition. Deposition has been a non-
negligible pathway that human pollution is contaminating the nearly untouched areas.

We calculate the ratio of $NO_y$ deposition to $NO_x$ emission (Fig. 6(c)). In continental non-

coastal regions, the average ratio is 74% (81% if taking consideration of coastal regions). In high
$NO_x$ emission regions (i.e. North America, East Asia and South Asia), an average 60-80% of the
$NO_y$ is removed by deposition, with large regional variation. For low emission regions (i.e. North
Africa and Central Asia), the ratio can reach higher than 90%. Also in coastal regions and open
ocean, the ratio is generally over 200%. Instead of the local emission, the transport of air
pollutants from elsewhere is the major source of deposition.
### 3.3.2   $NH_x$ Deposition
The global $NH_x$ deposition is 54 Tg(N) in 2010, with 69% of deposits on continental non-coastal
regions, 19% of deposits on ocean non-coastal regions and 13% of deposits on coastal regions
(Table 4). For continental non-coastal regions, South Asia receives 16% of global $NH_x$
depositions, followed by East Asia (13%). The whole Indian peninsula receives higher $NH_x$
depositions than 2,000 mg(N) m$^{-2}$ yr$^{-1}$ (Fig. 6(e)). Also, the Asian regions have several high
deposition areas: the North China Plain and Indonesia (1,200-2,000 mg(N) m$^{-2}$ yr$^{-1}$), Japan,
Thailand, Vietnam and Myanmar (500-600 mg(N) m$^{-2}$ yr$^{-1}$). Other regions with high $NH_x$
deposition are:  the United States Midwest, Germany, France, Northern Italy, Southern Brazil
and Ethiopia (400-800 mg(N) m$^{-2}$ yr$^{-1}$). Distributions of components of $NH_x$ deposition are
shown in Fig. S15 in the supplementary material.

Coastal regions of Southeast Asia (3%), East Asia (2%) and South Asia (2%) receive the

largest $NH_x$ deposition (~200-400 mg(N) m$^{-2}$ yr$^{-1}$). The east coast of North America and Mexico
also have high $NH_x$ deposition (150-200 mg(N) m$^{-2}$ yr$^{-1}$). Compared to $NO_y$ deposition, the $NH_x$
deposition on coastal regions is relatively lower. The ocean receives 17 Tg(N) of $NH_x$ deposition
in 2010, within the range of 13-29 Tg(N) estimated by Duce et al. (2008), but lower than 23.5
Tg(N) estimated by Dentener et al. (2006) and 21.4 Tg(N) estimated by Vet et al. (2014). 31% of
$NH_3$ emission is deposited on ocean areas, similar to 31% estimated by Dentener et al. (2006)
and 30% estimated by Lamarque et al. (2005), but slightly lower than 37% in PhotoComp
(Dentener et al., 2006) and 37% in HTAP I (Vet et al., 2014). The ocean emitted 12 Tg(N) of
$NH_3$ in 2010, which means that at least 5 Tg(N) of $NH_x$ deposition on oceans in 2010  came from
continental regions. This value is considerably lower than the 13 Tg(N) of deposition-emission
difference for $NO_y$ (including the 2 Tg(N) difference on coastal regions). A possible explanation
is that $NH_3$ has a short lifetime in the atmosphere, which makes it more likely to deposit close to
where it is emitted (Shen et al., 2016), while $NO_x$ can be oxidized to organic nitrate (Moxim et
al., 1996), which facilitates the long-range transport from land to open ocean.

We calculate the ratio of $NH_x$ deposition to $NH_3$ emission (Fig. 6(f)). The average ratio is

87% for continental non-coastal regions (92% if also considers the coastal regions). The ratios
are generally higher than those of $NO_y$ deposition (74% and 81%), since large a proportion of
$NH_x$ deposits near the source. The ratios are generally over 400% for coastal areas, but less than
100% on open ocean (70-90%). This is because there is less continental $NH_x$ transported to the
open ocean than to coastal regions.

### 3.3.3 N deposition

The global N deposition in 2010 is 113 Tg(N), with 65% of deposits on the continental non-coastal regions, 20% on non-coastal oceans and 15% on coastal regions (Table 4). East Asia (13%) and South Asia (11%) receive the largest amount of N deposition, consistent with the fact that they are also the largest N emission sources (16% and 13% respectively). The deposition reaches 3000 mg(N) m$^{-2}$ yr$^{-1}$ over Eastern China (especially North China Plain) and 2000 mg(N) m$^{-2}$ yr$^{-1}$ over India and Southeast Asia (Thailand, Vietnam and Malaysia). Other regions of high N deposition are the United States, northeast Western Europe (800-1200 mg(N) m$^{-2}$ yr$^{-1}$), Mexico, Central America, Brazil, northern Sub-Saharan Africa and the northeastern Middle East (500-600 mg(N) m$^{-2}$ yr$^{-1}$). For coastal regions, the east coast of the United States, all coasts of India and the east coast of East Asia are identified with relatively high deposition (>600 mg(N) m$^{-2}$ yr$^{-1}$).

Table 5 compares the N emission and deposition in HTAP II with HTAP I. The global N emission increases from 105 Tg(N) to 115 Tg(N), with a 12 Tg(N) (15%) increase in continental non-coastal regions and a 2 Tg(N) (14%) decrease in coastal regions. The change on the ocean is small due to increased NO$_y$ deposition but decreased NH$_x$ deposition. For continental non-coastal regions, increases in N emission are found in South Asia (5 Tg(N), 56%), East Asia (4 Tg(N), 26%) and Southeast Asia (2 Tg(N), 58%), while the emission in Europe decreases by 1 Tg(N) (12%). The emission changes in coastal regions are relatively small. The global N deposition increases by 7 Tg(N), with a 9 Tg(N) (14%) increase in continental non-coastal regions and a 2 Tg(N) decrease in ocean regions. Asian regions also have experienced the largest increases in deposition, and the amounts are identical with corresponding emission changes. Fig. S16 (b) compares the distribution of N deposition in HTAP II with HTAP I. Elevated N deposition is found in India, Indonesia and North Chain Plain (1,500 mg(N) m$^{-2}$ yr$^{-1}$). Regions with small increases are Japan, the northern Middle East, northwestern Brazil and Mexico (~200 mg(N) m$^{-2}$ yr$^{-1}$). On the other hand, the N deposition in the eastern United States and Europe have decreased by 200-400 mg(N) m$^{-2}$ yr$^{-1}$.

The global N dry and wet deposition is 40 Tg(N) yr$^{-1}$ and 73 Tg(N) yr$^{-1}$ in 2010, respectively. We calculate the ratio of dry deposition as $\frac{dry\ deposition}{dry\ deposition + wet\ deposition} \times 100\%$. For continental non-coastal regions, about 44% (range from 35-61%) of the N deposition comes from dry

deposition (42% if take coastal regions into consideration). If the overestimation of N dry
deposition in Section 3.1.2 is considered, this ratio could be even lower. Desert areas (e.g., the
Sonoran, Mojave and Chihuahuan deserts near the west coast of North America, the Sahara
Desert in North Africa, the Arabian Desert in Middle East and the Great Victoria Desert in
Australia) are seen with high ratios of dry deposition (80%) (Red color regions in Fig. 7(c)). This
outcome is reasonable since these areas generally lack precipitation. Low fractions of dry
deposition (30%) are found in Russia, Western China, Southeast Asia, Australia and Central
America. Almost all coastal regions are dominated by wet deposition. A study by Jickells (2006)
reported a dry deposition ratio of 21-45% for the east coast of the United States and a study by
Baker et al. (2010) suggested a ratio of 15-22% for the Atlantic Ocean. Our study receives
similar ratios for these coastal regions. A study by Bey et al. (2001) found an outflow of $NO_y$
from Asia over the Western Pacific Ocean through deposition. According to this study, about
70% of this land-to-ocean export of $NO_y$ deposition is through wet deposition (Fig.7 (a)).
The $NH_x$ and $NO_y$ deposition is 54 Tg(N) $yr^{-1}$ and 59 Tg(N) $yr^{-1}$ in 2010, respectively.
The average ratio of $NH_x$ deposition (calculated as $\frac{NH_x\ deposition}{NH_x\ deposition + NO_y\ deposition} \times 100\%$) for
continental non-coastal regions is 47% (45% if coastal regions are taken into consideration).
South Asia (71%) and Southeast Asia (63%) are dominated by $NH_x$ deposition, owing to high
local $NH_3$ emission, while the Middle East (25%) and North Africa (34%) are dominated by $NO_y$
deposition. Fig. 7(f) shows the global distribution of the ratio of $NH_x$ deposition. Except the high
ratio found in the Indian peninsula, Southeast Asia, Southeast Brazil, South Argentina and New
Zealand (70-80%) and Eastern Asia (~60%), other continental non-coastal regions are mainly
dominated by $NO_y$ deposition. This is consistent with finding by ACCMIP (Sun et al., 2016). We
compare the ratio of $NH_x$ deposition in 2010 (HTAP II) with that in 2001 (HTAP I) (Fig. S17).
Generally, we found a 10% worldwide decrease in the ratio of NHx deposition from 2001 to
2010. In particular, a 30% decrease is found in southeastern China, mainly due to the large
increase in $NO_x$ emission during the last decade. On the other hand, the ratio of $NH_x$ deposition
in California was 15-20% in 2001 and increases to 40-60% in 2010. The ratio in Alaska also
increases from 30-40% to 50%. There is a generally 5-10% increase over the eastern United
States. This is consistent with an observed large increase of the $NH_x$ depositions and decrease of
$NO_y$ depositions in northeastern United States from 1990s to 2010s (Du et al., 2014;Li et al.,
2016). A possible explanation is that the implementation of emission control stretegies such as
the Clean Air Act (CAA) has resulted in a large reduction in $NO_x$ emssions, which lowered the
$NO_y$ deposition in the United States (Lloret and Valiela, 2016). This benefit is compensated by
increasing $NH_x$ deposition because no limitation is implemented on $NH_3$ emission (Kanakidou et
al., 2016;Li et al., 2016). Some regions have small increases in the ratio of $NH_x$ deposition, such
as North Europe (Norway) (5%), Southeast Asia (10%) and Western Australia (10%).

## 4    Conclusions

We calculate the S and N deposition in 2010 using the multi-model mean (MMM) of an 11-
model ensemble from the HTAP II project. The model performance on wet deposition is
evaluated with measurement networks of NADP over North America, EMEP over Europe and
EANET over East Asia. The modelled wet deposition compares favorably with the observations.
About 76-83% of stations are predicted within ±50% of observations. $SO_4^{2-}$ wet deposition is
underestimated in East Asia by 20%, especially at 3 Chinese stations with high $Ca^{2+}$
concentration. Because the locations of the Chinese stations don't cover the areas with highest
deposition, it is hard to provide a comprehensive evaluation over this region. For $NO_3^-$ wet
deposition, 20% positive model bias is generally found at stations in eastern United States, while
some European (Poland, Norway and Spain) and East Asian (in Southeast Asia) stations with
high observed deposition are underestimated by about 60-70%. $NH_4^+$ wet deposition is
underestimated in Europe (especially in Norway and Poland) and East Asia (especially in Russia
and Korea). An inter-comparison is conducted with previous projects of PhotoComp, ACCMIP
and HTAP I. HTAP II has significantly improved the estimation of both S and N deposition at
European stations compared to that in previous projects. Improved estimates are also found in
East Asia. Modelled dry deposition is compared with the inferential data from CASTNET in
North America. The MMM results are generally higher than the inferential data by 50-170%,
which is also reported in ACCMIP and HTAP I studies.
We calculate the S and N depositions on lands, costal zones and open oceans. The global
S deposition is 84 Tg(S) in 2010, with 49% deposits on continental non-coastal regions, 32%
deposits on non-coastal oceans and 19% deposits on coastal regions. The global N deposition is
113 Tg(N) in 2010, of which 59 Tg(N) is $NO_y$ deposition and 64 Tg(N) is $NH_x$ deposition. 65%
of N is deposited on the continental non-coastal regions and 35% is on oceans (including 15% on
coastal regions). For continental regions, high S deposition is found in Asia regions (East Asia,
South Asia and Southeast Asia), United States Midwest, Central America and Eastern Europe.
For N deposition, high deposition is also identified in the above-mentioned regions plus the Sub
Sahara Africa and Brazil. For coastal regions, the east coast of Asia, all coasts of India and
Malaysia and east coast of Unites States are seen with relatively high S and N deposition.
According to our estimation, about 4 Tg(S) of S deposition and 18 Tg(N) of N deposition are
exported from land to ocean, including 0.3 Tg(S) and 4 Tg(N) in coastal regions.

We compare the HTAP II results in 2010 with HTAP I in 2001 by using the same land-
ocean mask. The S deposition decreases 2 Tg(S) from 2001 to 2010, with significant decreases in
Europe (5 Tg(S)), North America (3 Tg(S)) and Russia (2 Tg(S)), and increases in South Asia (2
Tg(S)) and the Middle East (1 Tg(S)). East Asia doesn't have large net changes in its S
deposition due to increased S emission from 2001-2005 and a continuous reduction in S emission
starting from 2006 owing to the $SO_2$ control policies in China's 11[th] FYP. The N deposition
increases by 7 Tg(N). The increased N emissions from South Asia (5 Tg(N)), East Asia (4
Tg(N)) and Southeast Asia (2 Tg(N)) lead to identical amounts of elevation in deposition in
corresponding regions. We also compare the ratio of $NH_x$ deposition in total N deposition
between HTAP I and HTAP II. The ratio has increased in some regions of North America,
especially in California (~20%), Alaska (~10%) and the eastern United States (5-10%), which
agrees well with recent observational and modelling studies in United States. A small increase in
the ratio of $NH_x$ deposition is found in North Europe (Norway) (5%), Southeast Asia (10%) and
Western Austrilia (10%). On the other hand, $NO_y$ deposition starts to dominate in East Asia
(especially China) due to increased $NO_x$ emission in recent years.

This study updates our knowledge about the global S and N deposition in 2010. We find
that the global distributions of S and N depositions have changed considerably during the last 10
years, with decreases in North America and Europe and increases in Asian regions. Further
studies could determine how much these changes could affect the source-receptor relationship on
deposition between continents and the impact of this relationship on global agriculture and
ecosystems?
*Acknowledgements*
We thank all participating modelling groups in HTAP II for providing the simulation data. We
thank Dr. Robert Vet for providing the multi-model ensemble results of HTAP I.  The National
Center for Atmospheric Research is sponsored by the National Science Foundation. The CESM

project is supported by the National Science Foundation and the Office of Science (BER) of the U.S. Department of Energy. Computing resources were provided by the Climate Simulation Laboratory at NCAR's Computational and Information Systems Laboratory (CISL), sponsored by the National Science Foundation and other agencies. We acknowledge the support by NASA HAQAST (grant no. NNX16AQ19G). We also acknowledge the support by Supercomputer system of the National Institute for Environmental Studies, Japan and The Environment Research and Technology Development Fund (S-12-3) of the Ministry of the Environment, Japan, JSPS KAKENHI (grant no. 5H01728).

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

**Figures**
Caption:
Fig. 1 Evaluation of MMM performance of $SO_4^{2-}$, $NO_3^-$ and $NH_4^+$ wet deposition (mg (N or S)
$m^{-2}$ $yr^{-1}$) at NADP (left), EMEP (middle) and EANET (right) stations. The MMM is the annual
wet deposition in 2010 and the observation is 3-year average annual data of 2009-2011.
Performances of individual models are in Fig. S2-S4 in the supplementary material.

Fig. 2. Distribution of $SO_4^{2-}$, $NO_3^-$ and $NH_4^+$ wet deposition (mg (N or S) $m^{-2}$ $yr^{-1}$) of MMM and
observation. The MMM is the annual wet deposition in 2010 and the observation is 3-year
average annual data of 2009-2011.Contours are MMM results and filled circles are observation.

Fig. 3 Evaluation of MMM performance of $SO_2$, $SO_4^{2-}$, $NO_3^-$, $HNO_3$ and $NH_4^+$ dry deposition
(mg (N or S) $m^{-2}$ $yr^{-1}$) at CASTNET stations. The MMM is the annual dry deposition in 2010 and
the observation data is 3-year average annual data during 2009-2011 from CASTNET network.
Performances of individual models are in Fig. S7-S11 in the supplementary material.

Fig. 4. Distribution of $SO_2$, $SO_4^{2-}$, $NO_3^-$, $HNO_3$ and $NH_4^+$ dry deposition (mg (N or S) $m^{-2}$ $yr^{-1}$) of
MMM and observation. The MMM is the annual dry deposition in 2010 and the observation is 3-
year average annual data of 2009-2011. Contours are MMM results and filled circles are
inferential data from CASTNET.

Fig. 5 (top panel) MMM results of S emission and deposition in 2010 (mg(S) $m^{-2}$ $yr^{-1}$) and ratio
of S deposition in S emission (%). (bottom panel) MMM results of S dry and wet deposition in
2010 (mg(S) $m^{-2}$ $yr^{-1}$) and ratio of dry deposition in total (wet+dry) deposition (%).

Fig. 6 MMM results of $NO_X$, $NH_3$ and $N(NO_X + NH_3)$ emission (mg(N) $m^{-2}$ $yr^{-1}$) (left panel),
$NO_y$, $NH_X$ and N ($NO_y+NH_X$) deposition (mg(N) $m^{-2}$ $yr^{-1}$) in 2010. (middel panel), ratio of $NO_y$,
$NH_X$ and N deposition to $NO_X$, $NH_3$ and $N(NO_X + NH_3)$ emission (%) (right panel). purple
colors represent regions where deposition is larger than emission.

Fig. 7 (top panel) The percentage of dry deposition in wet+dry deposition for $NO_y$, $NH_x$ and N
($NO_y+NH_x$) deposition. The ratio is calculated as (dry deposition)/ (dry+wet deposition) ×100%.
(bottom panel) The percentage of NHx deposition in N ($NO_y+NH_x$) deposition for wet, dry and
(wet+dry) deposition. The ratio is calculated as ($NH_x$ deposition)/ ($NO_y+NH_x$ deposition).





Fig. 1

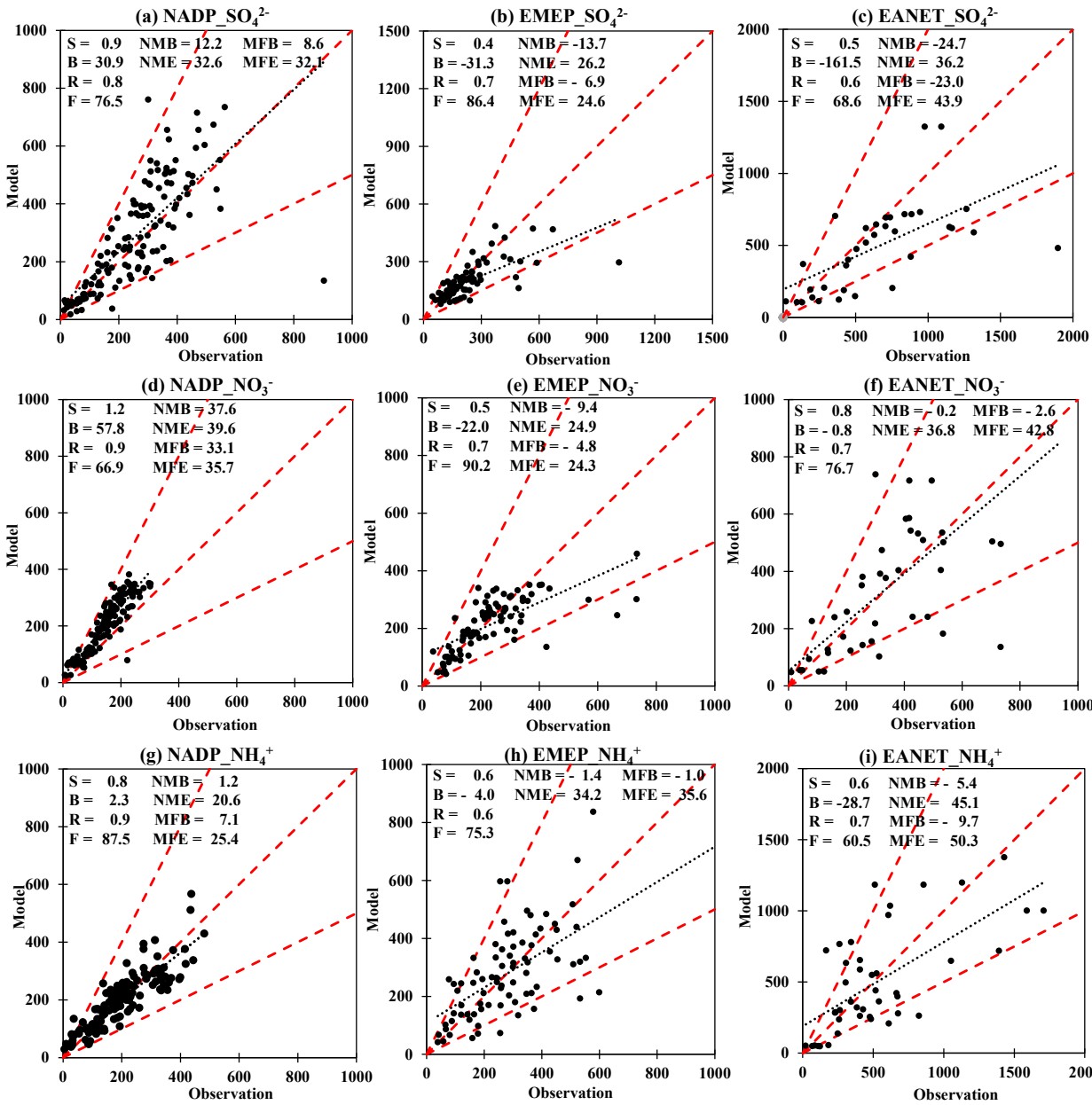

Fig. 1 Evaluation of MMM performance of $SO_4^{2-}$, $NO_3^-$ and $NH_4^+$ wet deposition (mg (N or S)
$m^{-2}$ $yr^{-1}$) at NADP (left), EMEP (middle) and EANET (right) stations. The MMM is the annual
wet deposition in 2010 and the observation is 3-year average annual data of 2009-2011.
Performances of individual models are in Fig. S2-S4 in the supplementary material.
Fig. 2

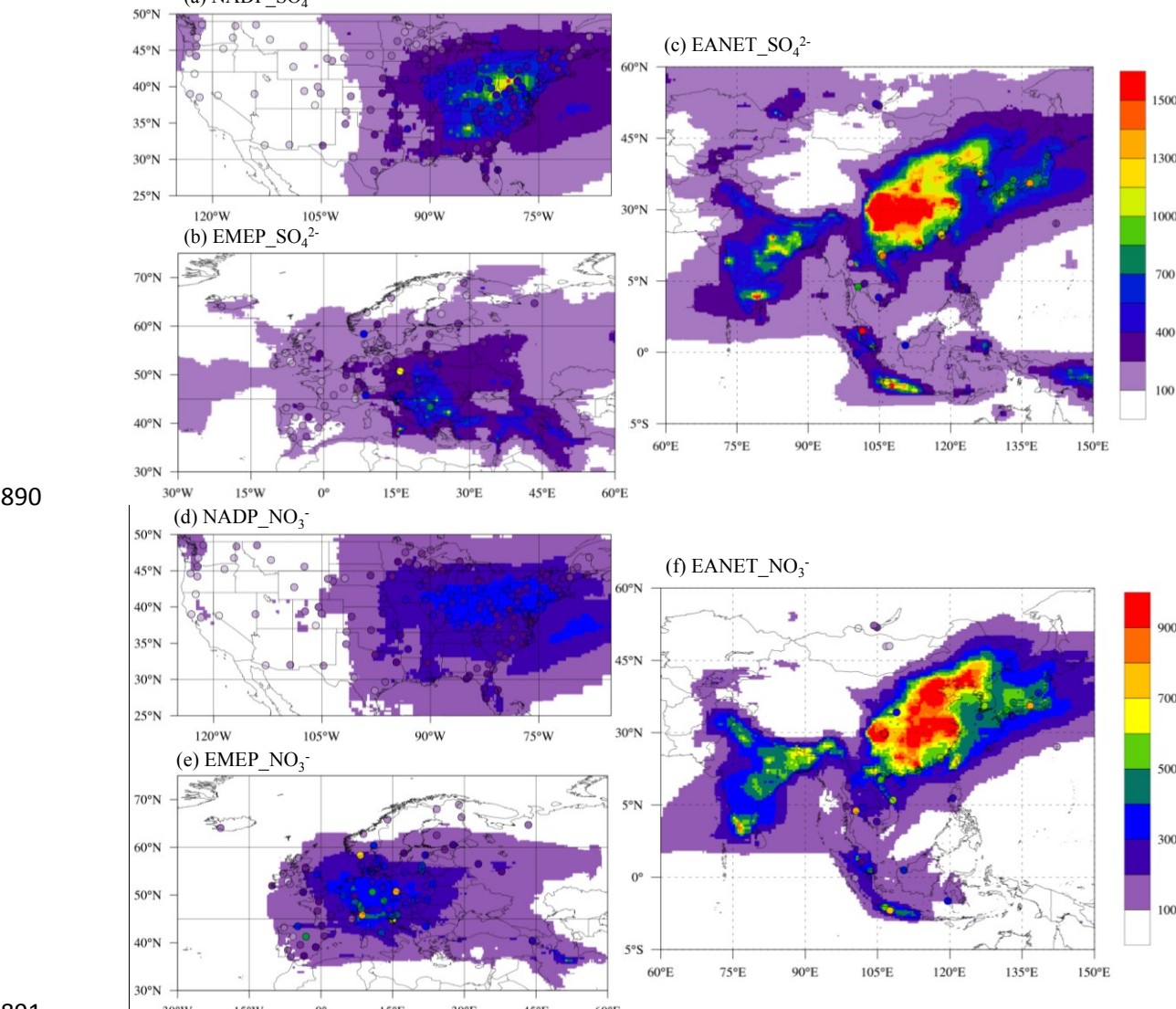



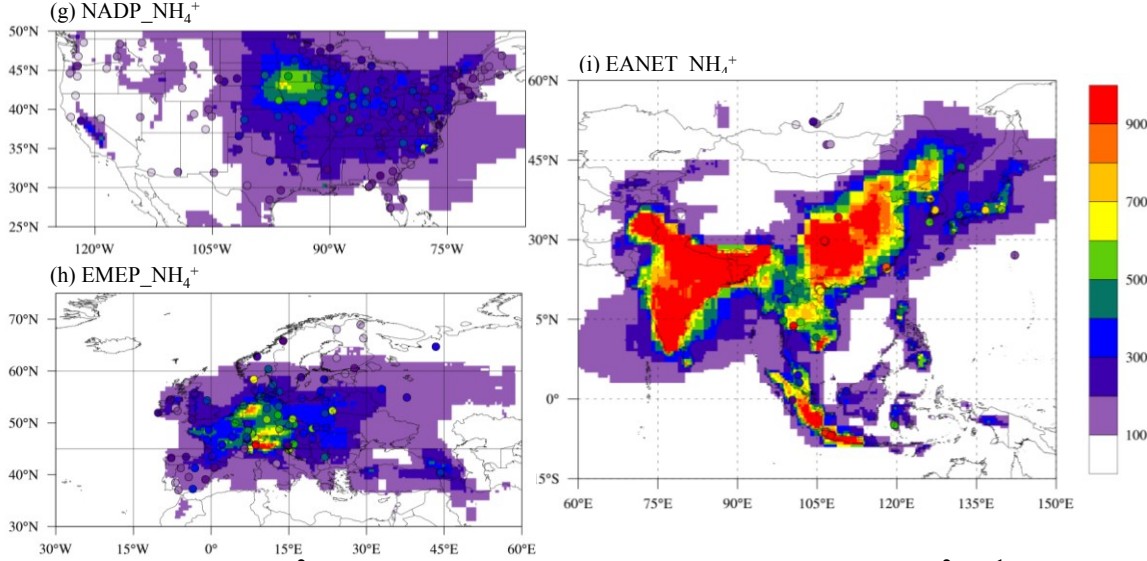

Fig. 2. Distribution of $SO_4^{2-}$, $NO_3^-$ and $NH_4^+$ wet deposition (mg (N or S) m$^{-2}$ yr$^{-1}$) of MMM and
observation. The MMM is the annual wet deposition in 2010 and the observation is 3-year
average annual data of 2009-2011.Contours are MMM results and filled circles are observation.

Fig. 3

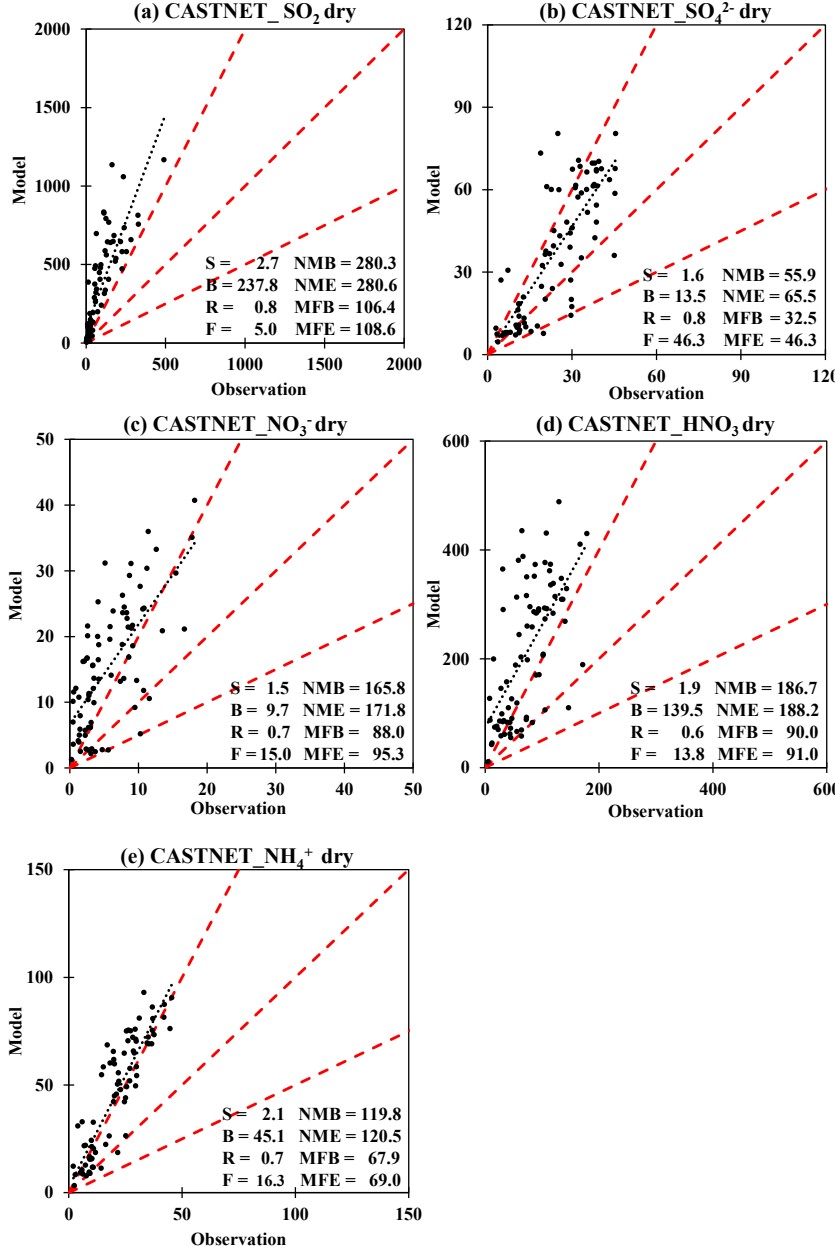

Fig. 3 Evaluation of MMM performance of SO₂, SO₄²⁻, NO₃⁻, HNO₃ and NH₄⁺ dry deposition
(mg (N or S) m⁻² yr⁻¹) at CASTNET stations. The MMM is the annual dry deposition in 2010 and
the observation data is 3-year average annual data during 2009-2011 from CASTNET network.
Performances of individual models are in Fig. S7-S11 in the supplementary material.

Fig. 4

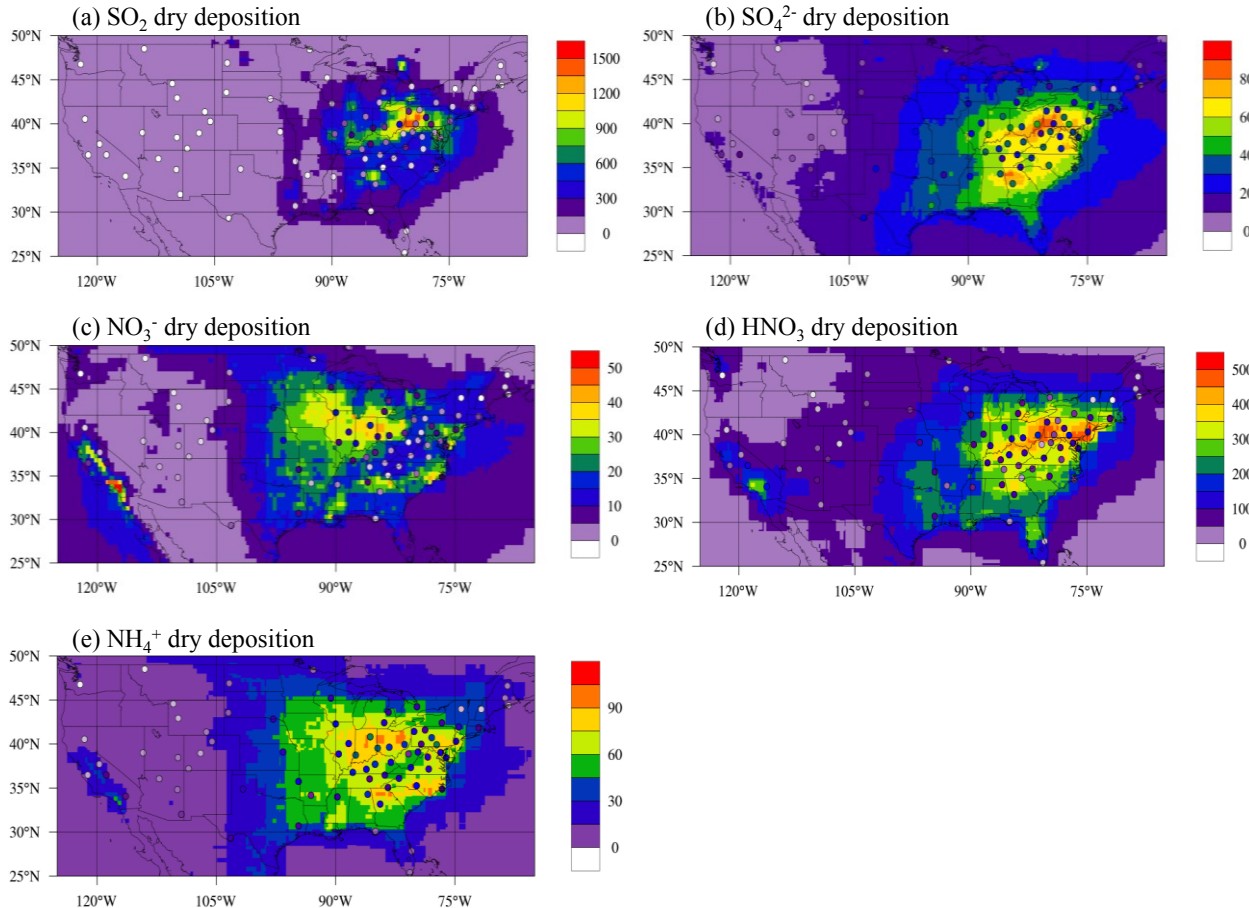

Fig. 4. Distribution of $SO_2$, $SO_4^{2-}$, $NO_3^-$, $HNO_3$ and $NH_4^+$ dry deposition (mg (N or S) $m^{-2}$ $yr^{-1}$) of
MMM and observation. The MMM is the annual dry deposition in 2010 and the observation is 3-
year average annual data of 2009-2011. Contours are MMM results and filled circles are
inferential data from CASTNET.

Fig. 5

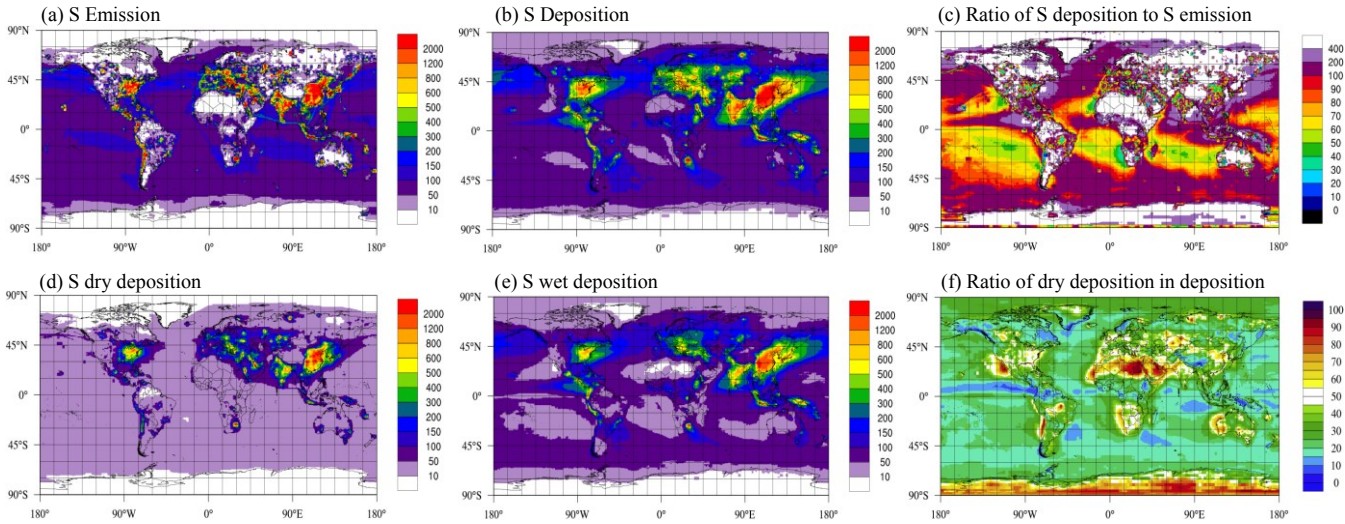

Fig. 5 (top panel) MMM results of S emission and deposition in 2010 (mg(S) m$^{-2}$ yr$^{-1}$) and ratio
of S deposition in S emission (%). (bottom panel) MMM results of S dry and wet deposition in
2010 (mg(S) m$^{-2}$ yr$^{-1}$) and ratio of dry deposition in total (wet+dry) deposition (%).

Fig. 6

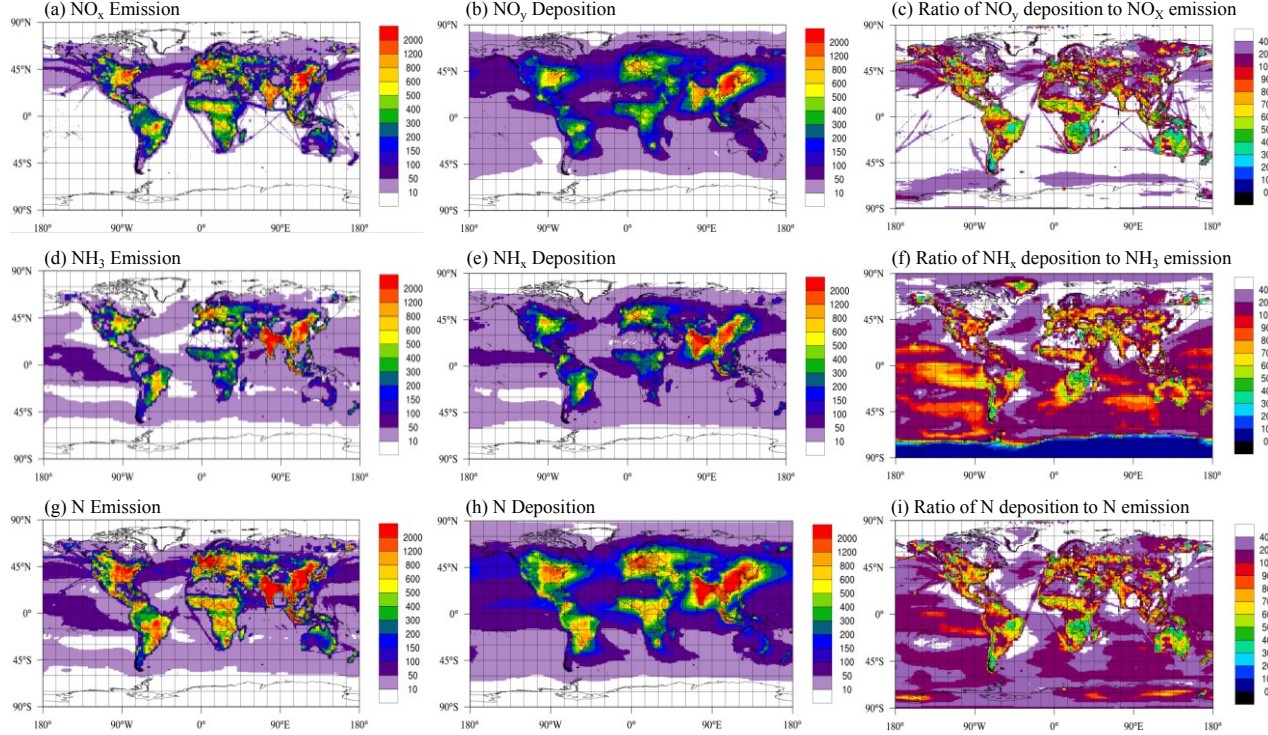

Fig. 6 MMM results of NO$_X$, NH$_3$ and N(NO$_X$ + NH$_3$) emission (mg(N) m$^{-2}$ yr$^{-1}$) (left panel),
NO$_y$, NH$_X$ and N (NO$_y$+NH$_X$) deposition (mg(N) m$^{-2}$ yr$^{-1}$) in 2010. (middel panel), ratio of NO$_y$,
NH$_X$ and N deposition to NO$_X$, NH$_3$ and N(NO$_X$ + NH$_3$) emission (%) (right panel). purple
colors represent regions where deposition is larger than emission.

Fig. 7

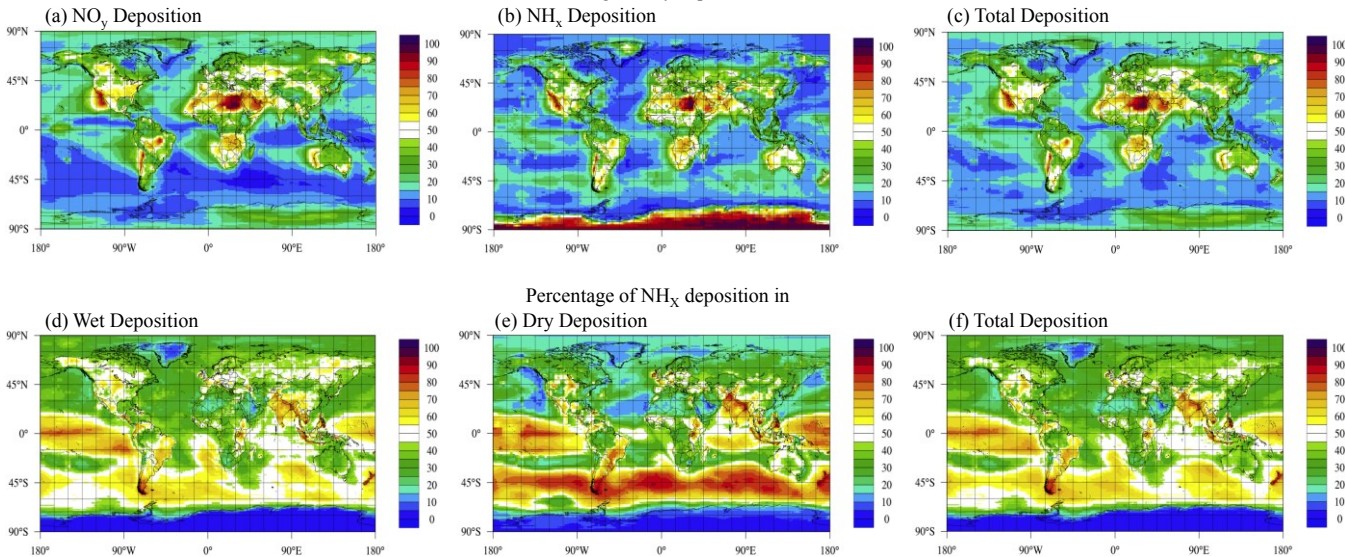

Fig. 7 (top panel) The percentage of dry deposition in wet+dry deposition for NO$_y$, NH$_x$ and N
(NO$_y$+NH$_x$) deposition. The ratio is calculated as (dry deposition)/ (dry+wet deposition) ×100%.
(bottom panel) The percentage of NH$_x$ deposition in N (NO$_y$+NH$_x$) deposition for wet, dry and
(wet+dry) deposition. The ratio is calculated as (NH$_x$ deposition)/ (NO$_y$+NH$_x$ deposition).

**Tables**

Table 1. Intercomparison of HTAP II MMM performance with previous projects on wet deposition. The unit is mg
(N or S) m$^{-2}$ yr$^{-1}$.

| Wet SO$_4^{2-}$ Deposition | North America | | | | Europe | | | | Asia | | | |
|---|---|---|---|---|---|---|---|---|---|---|---|---|
| | PhotoComp | HTAP I | ACCMIP | HTAP II | PhotoComp | HTAP I | ACCMIP | HTAP II | PhotoComp | HTAP I | ACCMIP | HTAP II |
| Linear Fit Slope | 0.9 | 1 | 0.6 | 0.9 | 0.4 | 0.6 | 0.3 | 0.4 | 0.4 | 0.5 | 0.3 | 0.5 |
| Mean Bias | 46.3 | 50 | -18.8 | 30.9 | -67.1 | 51.5 | -125.3 | -31.3 | -218.6 | -182.1 | -292.4 | -161.5 |
| Mean Observation | 309.8 | 309.8 | 309.8 | 253.7 | 404.5 | 404.5 | 404.5 | 228.7 | 686.1 | 686.1 | 686.1 | 653.7 |
| Mean Model | 356.1 | 359.8 | 291 | 284.6 | 337.3 | 456.1 | 279.3 | 197.4 | 467.5 | 504.1 | 393.7 | 492.2 |
| R | 0.9 | 0.9 | 0.9 | 0.8 | 0.6 | 0.6 | 0.6 | 0.7 | 0.9 | 0.9 | 0.8 | 0.6 |
| Fraction within ±50% | 70.4 | 70 | 72.2 | 76.5 | 78.7 | 52.8 | 78.7 | 86.4 | 80 | 88 | 72 | 68.6 |
| Number of stations | 346 | 346 | 346 | 136 | 126 | 126 | 126 | 82 | 49 | 49 | 49 | 43 |

| Wet NO$_3^-$ Deposition | North America | | | | Europe | | | | Asia | | | |
|---|---|---|---|---|---|---|---|---|---|---|---|---|
| | PhotoComp | HTAP I | ACCMIP | HTAP II | PhotoComp | HTAP I | ACCMIP | HTAP II | PhotoComp | HTAP I | ACCMIP | HTAP II |
| Linear Fit Slope | 1 | 1 | 0.9 | 1.2 | 0.3 | 0.3 | 0.3 | 0.5 | 0.5 | 0.5 | 0.4 | 0.8 |
| Mean Bias | 34.8 | 21.9 | 44.3 | 57.8 | -41.4 | -60 | -75.2 | -22.0 | -47.8 | -49.3 | -46.4 | -0.8 |
| Mean Observation | 191.3 | 191.3 | 191.3 | 153.7 | 300.5 | 300.5 | 300.5 | 237.3 | 263 | 263 | 263 | 356.4 |
| Mean Model | 226.1 | 213.3 | 235.6 | 211.5 | 259.1 | 240.5 | 225.3 | 215.4 | 215.2 | 213.7 | 216.7 | 355.7 |
| R | 0.8 | 0.9 | 0.9 | 0.9 | 0.6 | 0.6 | 0.6 | 0.7 | 0.8 | 0.8 | 0.8 | 0.7 |
| Fraction within ±50% | 77 | 84.3 | 68.7 | 66.9 | 75 | 85.2 | 85.2 | 90.2 | 84 | 84 | 88 | 76.7 |
| Number of stations | 346 | 346 | 346 | 136 | 126 | 126 | 126 | 82 | 49 | 49 | 49 | 43 |

| Wet NH$_4^+$ Deposition | North America | | | | Europe | | | | Asia | | | |
|---|---|---|---|---|---|---|---|---|---|---|---|---|
| | PhotoComp | HTAP I | ACCMIP | HTAP II | PhotoComp | HTAP I | ACCMIP | HTAP II | PhotoComp | HTAP I | ACCMIP | HTAP II |
| Linear Fit Slope | 0.8 | 0.9 | 0.5 | 0.8 | 0.4 | 0.4 | 0.3 | 0.6 | 0.8 | 0.7 | 0.1 | 0.6 |
| Mean Bias | 5.5 | 10.9 | -12.1 | 2.3 | -23.9 | -49.7 | -94.7 | -4.0 | -69.7 | -63.4 | -136.2 | -28.7 |
| Mean Observation | 161.3 | 161.3 | 161.3 | 195.5 | 336 | 336 | 336 | 286.1 | 400.5 | 400.5 | 400.5 | 534.5 |
| Mean Model | 166.8 | 172.2 | 149.2 | 197.9 | 312.1 | 286.4 | 241.3 | 282.2 | 330.8 | 337.1 | 264.4 | 505.8 |
| R | 0.9 | 0.9 | 0.8 | 0.9 | 0.8 | 0.6 | 0.6 | 0.6 | 0.8 | 0.8 | 0.2 | 0.7 |
| Fraction within ±50% | 82.2 | 84.8 | 75.7 | 87.5 | 73.9 | 79.5 | 78.4 | 75.3 | 76 | 68 | 56 | 60.5 |
| Number of stations | 346 | 346 | 346 | 136 | 126 | 126 | 126 | 82 | 49 | 49 | 49 | 43 |


Table 2. Intercomparison of HTAP II MMM performance with previous project on dry deposition. The unit is mg (N
or S) $m^{-2}$ $yr^{-1}$. S dry deposition is the sum of $SO_2$ and $SO_4^{2-}$ dry deposition. N dry deposition is the sum of $HNO_3$,
$NO_3^-$ and $NH_4^+$ dry deposition (not include $NO_2$ and $NH_3$ deposition).

| | S dry deposition | | | $SO_2$ dry deposition | | | $SO_4^{2-}$ dry deposition | | |
|---|---|---|---|---|---|---|---|---|---|
| | ACCMIP | HTAP I | HTAP II | ACCMIP | HTAP I | HTAP II | ACCMIP | HTAP I | HTAP II |
| Linear fit slope | 1 | - | 2.7 | 1 | - | 2.7 | 1 | - | 1.6 |
| Mean Bias | 280.9 | 367 | 251.2 | 264 | - | 237.8 | 17 | - | 13.5 |
| Mean observation | 225.6 | - | 108.9 | 191 | - | 84.8 | 35 | - | 24.1 |
| Mean model | 506.5 | - | 360.2 | 455 | - | 322.6 | 52 | - | 37.5 |
| R | 0.8 | 0.8 | 0.8 | 0.8 | - | 0.8 | 0.9 | - | 0.8 |
| Fraction within ±50% | 6 | - | 12.5 | 6 | - | 5 | 48 | - | 46.3 |

| | N dry deposition | | | $HNO_3$ dry deposition | | | $NH_4^+$ dry deposition | | |
|---|---|---|---|---|---|---|---|---|---|
| | ACCMIP | HTAP I | HTAP II | ACCMIP | HTAP I | HTAP II | ACCMIP | HTAP I | HTAP II |
| Linear fit slope | - | - | 2.1 | 1 | - | 1.9 | 2 | - | 2.1 |
| Mean Bias | - | 411 (eastern NA) 114 (western NA) | 185.1 | 75 | - | 139.5 | 33 | - | 24.6 |
| Mean observation | - | - | 101.1 | 119 | - | 74.7 | 28 | - | 20.5 |
| Mean model | - | - | 286.1 | 195 | - | 214.2 | 60 | - | 45.1 |
| R | - | 0.8 | 0.7 | 0.8 | - | 0.6 | 0.8 | - | 0.7 |
| Fraction within ±50% | - | - | 13.8 | 38 | - | 13.8 | 18 | - | 16.3 |


Table 3. MMM estimates of S deposition and emission in 2010 (Tg(S) yr$^{-1}$) and comparison with HTAP I results. Δ
is the difference between 2010 and 2001 calculated as (HTAP II – HTAP I). The number in parentheses is the
percentage of change, calculated as $\frac{(HTAP\ II\ -HTAP\ I)}{HTAP\ I} \times 100\%$.

| Regions | S emission | | | | | | S deposition | | | | | |
|---|---|---|---|---|---|---|---|---|---|---|---|---|
| | Non-coastal | | | Coastal | | | Non-coastal | | | Coastal | | |
| | HTAP II (2010) | HTAP I (2001) | Δ | HTAP II (2010) | HTAP I (2001) | Δ | HTAP II (2010) | HTAP I (2001) | Δ | HTAP II (2010) | HTAP I (2001) | Δ |
| 3. North America | 6.2 | 9.5 | -3.3 (-34.3) | 1.0 | 1.3 | -0.2 (-19.2) | 4.7 | 7.2 | -2.5 (-34.8) | 1.3 | 1.3 | 0.0 (-1.2) |
| 4. Europe | 3.9 | 10.0 | -6.1 (-60.8) | 1.6 | 3.6 | -1.9 (-54.2) | 2.7 | 6.4 | -3.7 (-58.2) | 1.5 | 2.9 | -1.4 (-49.6) |
| 5. South Asia | 5.2 | 3.3 | 1.9 (56.4) | 0.8 | 0.8 | 0.0 (-3.6) | 3.7 | 2.4 | 1.4 (57.8) | 1.0 | 0.9 | 0.1 (17.0) |
| 6. East Asia | 15.0 | 15.6 | -0.6 (-4.0) | 1.8 | 3.2 | -1.4 (-42.8) | 11.2 | 11.9 | -0.7 (-5.6) | 2.9 | 3.3 | -0.4 (-13.3) |
| 7. Southeast Asia | 2.5 | 1.7 | 0.7 (42.4) | 2.6 | 2.4 | 0.1 (6.0) | 2.4 | 1.9 | 0.5 (27.6) | 2.8 | 2.4 | 0.4 (16.1) |
| 8. Australia | 1.5 | 1.0 | 0.5 (56.0) | 2.0 | 1.4 | 0.6 (42.0) | 1.0 | 0.7 | 0.3 (43.9) | 1.5 | 1.1 | 0.3 (28.0) |
| 9. North Africa | 0.7 | 1.1 | -0.4 (-37.0) | 0.9 | 0.9 | 0.0 (-2.9) | 1.0 | 1.1 | -0.1 (-12.3) | 0.5 | 0.6 | -0.1 (-11.3) |
| 10. Sub Saharan Africa | 2.5 | 2.8 | -0.4 (-12.6) | 0.9 | 0.7 | 0.2 (24.2) | 2.7 | 2.6 | 0.1 (4.8) | 0.7 | 0.7 | 0.0 (-4.9) |
| 11. Middle East | 3.2 | 1.9 | 1.3 (68.9) | 1.1 | 0.5 | 0.6 (108.1) | 1.7 | 1.2 | 0.5 (47.0) | 0.6 | 0.4 | 0.2 (50.4) |
| 12. Central America | 2.2 | 2.1 | 0.2 (7.7) | 1.4 | 1.7 | -0.3 (-15.2) | 1.4 | 1.4 | 0.0 (1.6) | 1.4 | 1.4 | 0.0 (2.0) |
| 13. South America | 3.1 | 2.7 | 0.4 (16.9) | 0.8 | 1.0 | -0.2 (-23.3) | 2.4 | 2.1 | 0.3 (14.3) | 0.6 | 0.6 | 0.0 (1.6) |
| 14. RBU | 2.9 | 5.1 | -2.2 (-43.9) | 0.5 | 0.5 | 0.0 (-5.8) | 3.6 | 5.3 | -1.7 (-32.1) | 0.9 | 0.8 | 0.1 (9.7) |
| 15. Central Asia | 1.6 | 1.4 | 0.2 (18.3) | 0.0 | 0.0 | 0.0 (-5.9) | 1.2 | 1.2 | 0.0 (2.7) | 0.1 | 0.1 | 0.0 (-13.5) |
| 17. Antarctic | 1.1 | 1.1 | -0.1 (-7.2) | 0.0 | 0.0 | 0.0 (0) | 1.4 | 0.8 | 0.6 (73.7) | 0.0 | 0.0 | 0.0 (0) |
| Continental | 51.5 | 59.3 | -7.7 (-13.1) | 15.3 | 18.0 | -2.7 (-14.8) | 41.0 | 46.0 | -4.9 (-10.7) | 15.6 | 16.5 | -0.8 (-5.1) |
| 2. Ocean | 23.9 | 18.1 | 5.8 (31.9) | | | | 26.9 | 23.3 | 3.6 (15.5) | | | |
| 1. World Total | 75.4 | 77.4 | -2.0 (-2.6) | 15.3 | 18.0 | -2.7 (-14.8) | 67.9 | 69.2 | -1.3 (-1.9) | 15.6 | 16.5 | -0.8 (-5.1) |



Table 4. MMM estimates of N, $NO_y$ and $NH_X$ deposition and emission in 2010 (Tg(N) yr$^{-1}$) . The number in the
parenthesis is the percentage in world total emission/deposition.

| Regions | $NO_X$ emission | | $NO_y$ deposition | | $NH_3$ emission | | $NH_X$ deposition | | N emission | | N deposition | |
|---|---|---|---|---|---|---|---|---|---|---|---|---|
| | Non-coastal | Coastal | Non-coastal | Coastal | Non-coastal | Coastal | Non-coastal | Coastal | Non-coastal | Coastal | Non-coastal | Coastal |
| 3. North America | 6.6 (10.9) | 0.6 (1.1) | 4.4 (7.5) | 0.8 (1.4) | 3.7 (6.9) | 0.2 (0.3) | 3.4 (6.3) | 0.4 (0.7) | 10.3 (9.0) | 0.8 (0.7) | 7.8 (6.9) | 1.2 (1.0) |
| 4. Europe | 3.7 (6.2) | 1.2 (1.9) | 2.6 (4.4) | 1.2 (2.1) | 3.2 (5.9) | 0.6 (1.1) | 2.5 (4.6) | 0.8 (1.4) | 6.9 (6.0) | 1.8 (1.6) | 5.1 (4.5) | 2.0 (1.8) |
| 5. South Asia | 4.4 (7.3) | 0.4 (0.7) | 3.6 (6.0) | 0.7 (1.2) | 10.4 (19.2) | 0.7 (1.3) | 8.6 (15.9) | 1.0 (1.9) | 14.8 (12.9) | 1.1 (1.0) | 12.1 (10.7) | 1.7 (1.5) |
| 6. East Asia | 10.1 (16.8) | 1.3 (2.1) | 8.3 (14.0) | 2.2 (3.7) | 7.8 (14.4) | 0.7 (1.3) | 6.7 (12.5) | 1.0 (1.9) | 18.0 (15.7) | 2.0 (1.7) | 15.1 (13.3) | 3.2 (2.8) |
| 7. Southeast Asia | 2.6 (4.4) | 1.3 (2.1) | 1.9 (3.1) | 1.4 (2.3) | 3.1 (5.8) | 1.5 (2.7) | 3.2 (5.9) | 1.6 (2.9) | 5.8 (5.0) | 2.7 (2.4) | 5. 1 (4.5) | 2.9 (2.6) |
| 8. Australia | 1.4 (2.3) | 0.3 (0.6) | 0.6 (1.0) | 0.4 (0.7) | 0.7 (1.2) | 0.4 (0.8) | 0.4 (0.8) | 0.4 (0.8) | 2.0 (1.8) | 0.8 (0.7) | 1.0 (0.9) | 0.9 (0.8) |
| 9. North Africa | 1.5 (2.5) | 0.4 (0.7) | 1.4 (2.3) | 0.4 (0.6) | 0.9 (1.7) | 0.2 (0.3) | 0.7 (1.3) | 0.2 (0.3) | 2.5 (2.1) | 0.6 (0.5) | 2.1 (1.9) | 0.5 (0.5) |
| 10. Sub Saharan Africa | 7.4 (12.2) | 0.4 (0.7) | 4.7 (7.9) | 0.6 (1.1) | 4.0 (7.5) | 0.3 (0.6) | 3.4 (6.4) | 0.4 (0.7) | 11.4 (10.0) | 0.7 (0.6) | 8.1 (7.2) | 1.0 (0.9) |
| 11. Middle East | 1.9 (3.1) | 0.5 (0.7) | 1.4 (2.4) | 0.3 (0.6) | 0.7 (1.2) | 0.1 (0.2) | 0.5 (0.9) | 0.1 (0.2) | 2.5 (2.2) | 0.6 (0.5) | 1.9 (1.7) | 0.5 (0.4) |
| 12. Central America | 2.1 (3.5) | 0.8 (1.3) | 1.2 (2.1) | 0.8 (1.4) | 1.4 (2.6) | 0.5 (0.9) | 1.4 (2.5) | 0.6 (1.1) | 3.5 (3.1) | 1.2 (1.1) | 2.6 (2.3) | 1.4 (1.3) |
| 13. South America | 5.4 (8.9) | 0.3 (0.5) | 3.4 (5.8) | 0.3 (0.5) | 4.4 (8.1) | 0.3 (0.5) | 3.8 (7.1) | 0.3 (0.6) | 9.8 (8.5) | 0.6 (0.5) | 7.3 (6.4) | 0.6 (0.5) |
| 14. RBU | 2.4 (4.1) | 0.2 (0.3) | 2.4 (4.1) | 0.5 (0.9) | 1.7 (3.1) | 0.1 (0.2) | 1.8 (3.4) | 0.3 (0.6) | 4.1 (3.6) | 0.3 (0.2) | 4.3 (3.8) | 0.8 (0.7) |
| 15. Central Asia | 0.7 (1.1) | 0.0 (0) | 0.6 (1.1) | 0.0 (0.1) | 0.5 (0.9) | 0.0 (0) | 0.5 (0.8) | 0.0 (0) | 1.1 (1.0) | 0.0 (0) | 1.1 (1.0) | 0.1 (0.1) |
| 17. Antarctic | 0.0 (0.1) | 0.0 (0) | 0.1 (0.2) | 0.0 (0) | 0.0 (0.1) | 0.0 (0) | 0.1 (0.2) | 0.0 (0) | 0.1 (0.1) | 0.0 (0) | 0.2 (0.2) | 0.0 (0) |
| Continental | 50.2 (83.2) | 7.7 (12.8) | 36.7 (61.9) | 9.7 (16.4) | 42.6 (78.5) | 5.6 (10.3) | 37.0 (68.6) | 7.1 (13.1) | 92.9 (81.0) | 13.3 (11.6) | 73.7 (65.1) | 16.8 (14.8) |
| 2. Ocean | 2.4 (4) | | 12.9 (21.7) | | 6.0 (11.1) | | 9.9 (18.3) | | 8.5 (7.4) | | 22.8 (20.1) | |
| 1. World Total | 52.7 (87.2) | 7.7 (12.8) | 49.6 (83.6) | 9.7 (16.4) | 48.7 (89.7) | 5.6 (10.3) | 46.9 (86.9) | 7.1 (13.1) | 101.3 (88.4) | 13.3 (11.6) | 96.5 (85.2) | 16.8 (14.8) |


Table 5. Comparison of N deposition and emission between 2010 (HTAP II) and 2001 (HTAP I) (Tg (N) yr$^{-1}$). Δ is
the difference between 2010 and 2001 calculated as (HTAP II − HTAP I). The numbers in parentheses are the
percentage of change, calculated as $\frac{(HTAP\ II - HTAP\ I)}{HTAP\ I} \times 100\%$.

| Regions | N emission | | | | | | N deposition | | | | | |
|---|---|---|---|---|---|---|---|---|---|---|---|---|
| | Non-coastal | | | Coastal | | | Non-coastal | | | Coastal | | |
| | HTAP II (2010) | HTAP I (2001) | Δ | HTAP II (2010) | HTAP I (2001) | Δ | HTAP II (2010) | HTAP I (2001) | Δ | HTAP II (2010) | HTAP I (2001) | Δ |
| 3. North America | 10.3 | 10.2 | 0.1 (0.5) | 0.8 | 1.0 | -0.2 (-16.8) | 7.8 | 8.1 | -0.2 (-3.1) | 1.2 | 1.2 | -0.1 (-4.8) |
| 4. Europe | 6.9 | 7.8 | -0.9 (-11.8) | 1.8 | 2.7 | -0.9 (-33.6) | 5.1 | 5.7 | -0.7 (-11.4) | 2.0 | 2.6 | -0.6 (-23.6) |
| 5. South Asia | 14.8 | 9.5 | 5.3 (56.0) | 1.1 | 1.3 | -0.2 (-15.5) | 12.1 | 6.7 | 5.4 (79.7) | 1.7 | 1.7 | 0.1 (3.8) |
| 6. East Asia | 18.0 | 14.3 | 3.7 (25.9) | 2.0 | 2.2 | -0.2 (-8.1) | 15.1 | 11.9 | 3.2 (26.8) | 3.2 | 2.6 | 0.6 (21.9) |
| 7. Southeast Asia | 5.8 | 3.7 | 2.1 (57.4) | 2.7 | 2.7 | 0.0 (0.5) | 5.1 | 3.3 | 1.8 (54.4) | 2.9 | 3.0 | 0.0 (-0.7) |
| 8. Australia | 2.0 | 2.1 | -0.1 (-5.3) | 0.8 | 0.9 | -0.2 (-16.6) | 1.0 | 1.3 | -0.3 (-23.0) | 0.9 | 1.1 | -0.2 (-21.0) |
| 9. North Africa | 2.5 | 2.1 | 0.3 (15.6) | 0.6 | 0.6 | 0.1 (9.6) | 2.1 | 2.0 | 0.1 (7.5) | 0.5 | 0.6 | -0.1 (-12.2) |
| 10. Sub Saharan Africa | 11.4 | 11.8 | -0.4 (-3.1) | 0.7 | 1.1 | -0.3 (-30.6) | 8.1 | 9.1 | -1.0 (-10.9) | 1.0 | 1.5 | -0.4 (-30.2) |
| 11. Middle East | 2.5 | 1.8 | 0.8 (44.7) | 0.6 | 0.4 | 0.2 (36.8) | 1.9 | 1.4 | 0.5 (37.3) | 0.5 | 0.5 | 0.0 (0.2) |
| 12. Central America | 3.5 | 3.2 | 0.3 (9.6) | 1.2 | 1.5 | -0.2 (-16.5) | 2.6 | 2.4 | 0.2 (8.3) | 1.4 | 1.6 | -0.2 (-12.7) |
| 13. South America | 9.8 | 8.6 | 1.1 (12.8) | 0.6 | 0.8 | -0.2 (-23.4) | 7.3 | 6.8 | 0.5 (7.0) | 0.6 | 0.8 | -0.2 (-27.9) |
| 14. RBU | 4.1 | 4.7 | -0.6 (-12.4) | 0.3 | 0.3 | -0.1 (-17.4) | 4.3 | 4.9 | -0.6 (-12.6) | 0.8 | 0.7 | 0.1 (20.9) |
| 15. Central Asia | 1.1 | 1.1 | 0.0 (4.1) | 0.0 | 0.0 | 0.0 (24.5) | 1.1 | 1.2 | -0.1 (-5.1) | 0.1 | 0.1 | 0.0 (0) |
| 17. Antarctic | 0.1 | 0.1 | 0.0 (-17.5) | 0.0 | 0.0 | 0.0 (0) | 0.2 | 0.2 | 0.0 (-10.3) | 0.0 | 0.0 | 0.0 (0) |
| Continental | 92.9 | 81.1 | 11.8 (14.5) | 13.3 | 15.5 | -2.2 (-14.1) | 73.7 | 64.9 | 8.8 (13.5) | 16.8 | 17.9 | -1.1 (-6.1) |
| 2. Ocean | 8.5 | 8.4 | 0.0 (0.2) | | | | 22.8 | 23.5 | -0.7 (-2.9) | | | |
| 1. World Total | 101.3 | 89.6 | 11.8 (13.1) | 13.3 | 15.5 | -2.2 (-14.1) | 96.5 | 88.4 | 8.1 (9.2) | 16.8 | 17.9 | -1.1 (-6.1) |


Continue Table 5.

| | NO$_X$ emission | | NO$_y$ deposition | | NH$_3$ emission | | NH$_X$ deposition | |
|---|---|---|---|---|---|---|---|---|
| | Non-coastal | Coastal | Non-coastal | Coastal | Non-coastal | Coastal | Non-coastal | Coastal |
| | Δ | Δ | Δ | Δ | Δ | Δ | Δ | Δ |
| 3. North America | -0.1 | -0.1 | -0.4 | -0.1 | 0.1 | 0.0 | 0.1 | 0.0 |
| 4. Europe | -0.4 | -0.5 | -0.3 | -0.3 | -0.6 | -0.4 | -0.4 | -0.3 |
| 5. South Asia | 2.4 | 0.0 | 2.1 | 0.2 | 3.0 | -0.2 | 3.3 | -0.1 |
| 6. East Asia | 5.3 | 0.0 | 4.5 | 0.8 | -1.6 | -0.2 | -1.3 | -0.2 |
| 7. Southeast Asia | 1.1 | 0.1 | 0.7 | 0.1 | 1.0 | -0.1 | 1.1 | -0.2 |
| 8. Australia | 0.2 | 0.0 | -0.1 | 0.0 | -0.3 | -0.1 | -0.2 | -0.2 |
| 9 North Africa | 0.6 | 0.1 | 0.3 | 0.0 | -0.2 | 0.0 | -0.1 | -0.1 |
| 10. Sub Saharan Africa | 1.1 | -0.1 | 0.3 | -0.1 | -1.5 | -0.2 | -1.3 | -0.4 |
| 11 Middle East | 0.9 | 0.2 | 0.6 | 0.1 | -0.1 | 0.0 | -0.1 | -0.1 |
| 12. Central America | 0.6 | 0.0 | 0.2 | 0.0 | -0.3 | -0.2 | 0.0 | -0.2 |
| 13. South America | 1.5 | 0.0 | 0.7 | 0.0 | -0.3 | -0.2 | -0.3 | -0.2 |
| 14. RBU | 0.1 | 0.0 | 0.1 | 0.2 | -0.7 | -0.1 | -0.7 | 0.0 |
| 15. Central Asia | 0.2 | 0.0 | 0.1 | 0.0 | -0.1 | 0.0 | -0.1 | 0.0 |
| 17. Antarctic | 0.0 | 0.0 | 0.0 | 0.0 | 0.0 | 0.0 | -0.1 | 0.0 |
| Continental | 13.5 | -0.4 | 8.9 | 0.9 | -1.7 | -1.8 | -0.1 | -2.0 |
| 2. Ocean | 0.7 | | 1.7 | | -0.7 | | -2.4 | |
| 1. World Total | 14.2 | -0.4 | 10.7 | 0.9 | -2.4 | -1.8 | -2.6 | -2.0 |
