# Peer review of "Multi-model study of HTAP II on sulphur and nitrogen deposition"

_Atmospheric Chemistry and Physics, 2017_

## Referee Comment (RC1) · Anonymous Referee #2 · 29 Jan 2018

This study gives a comprehensive overview of the global atmospheric deposition of sulfur and nitrogen using a range of global atmospheric transport model, compared to observations for 2010. The manuscript is well written. I have a few questions and remarks:

Line 157. What does it mean that models are excluded if they fall outside their emission values? Several of the models given in table S1 and S2 don't calculate wet/total deposition? Are these models not used or have you deleted part of the calculations (i.e only used aerosol and not wet deposition) ? Maybe indicate which models used for each ensemble mean. If that is S1 and S2, maybe indicate what has been deleted? Are you looking at the surface emissions or total emissions, several models do not include emissions of DMS? A follow up question on this topic, why don't the total emissions

and deposition match up (i.e.7 Tg S and 1 Tg N differences in table 3 and 4)? Where does the left offers go, Have the models included organic N and S species?

Line 201. Data from 43 stations of the 52 available EANET stations are used. It seems like you have included all station times, urban as well as remote, which surely have different representativity for the region. Later you state that you delete sites with high Ca values (line 219), which can be an indication of urban dust, but these may also be from also from regional dust. Not sure if I understand the reasoning behind this way of selecting the sites.

Line215. The outliers in Norway and Poland are probably due these specific location with high precipitation amount (Norway) and high altitude (Polish site PL03 is at 1600moh.). Have you checked how well the models compare with precipitation amount contra concentration levels in precipitation?

Line 235 "According to Fig. 2(d), the over-predicted stations are mainly located in Midwestern and Southeast United". For me it seems like a general tendency (fig 1d). Maybe include information that 67 % of the station are within 50%.

Line 265. "The NH4 wet deposition is somewhat underestimated in all 3 regions". This is not the case for US (NADP) if one look at table 1 where the HTAPII is higher than observations

Line 350. "The ocean serves as an important sink of S deposition". But it is also a very important source. The net effect is only 3 TgS

Table 3,4,5. It is a bit confusing for the reader when you have defined two different categories continental coastal and ocean coastal which are the same thing. Would be more readable and less confusion if these cells are merges so it is clear that there are three categories (Ocean, Continent and Coast)

---

## Referee Comment (RC2) · Anonymous Referee #1 · 31 Jan 2018

Review comments for acp-2017-1121

This paper describes the global S and N emissions and deposition in a set of global models run for 2010 under a model intercomparison project (HTAP II) and compares the results with regional monitoring data and previous global modelling. The subject matter is of interest to air quality and ecosystem scientists, particularly those concerned with deposition to oceans, where there is little information from measurements or regional models. I recommend publication with minor revisions, with the most important being an expansion of the dry deposition discussion.

Specific comments:

Section 2.1: It isn't crucial, but it would be helpful to include a brief mention of any other

papers (published or forthcoming) that describe additional results from this intercomparison project (e.g. ambient concentrations of particulate matter or O3). With virtual special issues, it's not always obvious where to find related papers.

Section 2.2: Can you explain why some values in Tables S1-S3 seem inconsistent? E.g., total S emission from the MMM is 91 Tg in Table S1, but from equation (2) it seems that it should be 55+1+27=83 Tg. Is this an error or am I missing something? Also, the same table seems to show that OsloCTM3 should be excluded for S based on the mass balance criteria described. It's unclear why it is kept. I'm assuming model values that did not meet the criteria are not listed in the tables, though I don't think that was explicitly stated.

l. 265: the 81% value in the text does not match Table 1, which says 61%. Which is correct?

ll. 267-291 and Table 1: If possible, I suggest adding the number of stations for each comparison to Table 1 since that is likely different as well; where N is relatively low, the number and location of stations used could have a significant impact on the statistics.

Section 3.1.2: Additional discussion of dry deposition is warranted, given the large differences with the CASTNET inferential values. The difference between the CASTNET dry deposition calculations and those using the CAPMoN method are touched on (ll. 300-303) but the implications for the models is not fleshed out. I recommend moving this discussion to the end of the section 3.1.2 and discussing the relevance to the ensemble-measurement comparison. How does the CASTNET dry deposition velocity parametrization compare with those used in the various models? How do the modelled air concentrations of SO2, HNO3, etc. compare with the CASTNET observations?

l. 347: Australia appears to receive higher coastal S deposition than E. Asia, so should be listed here as well.

l. 374: Why the 32% increase in ocean S emissions? Is that real or the result of

improved emission budgets?

ll. 520-536: There is discussion of the areas of increasing NHx ratio, but globally there appears to be a general decrease (e.g. over the oceans). Maybe add a comment on this.

Fig. 2: Observation (point) values are very difficult to see on these small plots. Can they be enlarged, since the discussion in 3.1.1 hinges on the regional comparison? Fig. 4 is better; I would suggest that size is the minimum needed.

Technical comments:

The manuscript would generally benefit from careful copyediting to correct minor issues with non-standard English usage. I've only highlighted errors where the meaning was somewhat unclear:

l. 53: change "shows that. . . increases" to "predicts that. . . will increase"

ll. 67-90: previous results should all be in past tense

l. 123: The HTAP project? Task Force?

l. 130 and 146: update the Galmarini reference to the final ACP paper (2017)

l. 228: keeping with your sign convention in Table 1, the bias increases (or changes) from -160 to -300

l. 230 change "highest deposition" to "highest modelled deposition" if that is what is meant

ll. 240-243: Reword; the stations do not underestimate/under-predict the deposition, the MMM underestimates deposition at those stations.

l. 300: Reword to "Schwede et al. (2011) compared CASTNET dry deposition estimates with those of the Canadian. . ."

l. 312: suggest changing "0.5-1 times" to "50-100%" for consistency

ll. 321-22: change end of sentence to "...but this gradient is much weaker in the inferential data."

Suggest changing title of 3.2 to "Total S deposition" and similar for 3.3

l. 351-352: change to "...S deposition to the ocean and coastal areas in 2010." Remove text in parentheses.

l. 449: remove "and Mexico" since it's part of N. America

l. 485: replace "positive changes" with "increases" to avoid the message that this is a desirable change

l. 571: "large net changes" could replace "large changes" for clarity

Tables 3-5: Merge the coastal numbers into a single cell. Add text to the caption to remind the reader that the values in parentheses are percentages (Tables 3 and 4).

[Figure]

---

## Author Comment (AC1) · 9 Apr 2018

The responses to Referee #1: Please read the supplement file including responses to referee's comments that is referring to manuscript and supplement file that including additional figures and tables.

Please also note the supplement to this comment:
https://www.atmos-chem-phys-discuss.net/acp-2017-1121/acp-2017-1121-AC1-supplement.zip

---

## Author Comment (AC2) · 9 Apr 2018

Please review the supplement files including responses referee's comments, revised manuscript and supplement including additional figures and tables.

Please also note the supplement to this comment:
https://www.atmos-chem-phys-discuss.net/acp-2017-1121/acp-2017-1121-AC2-supplement.zip

---

## Author Response (AR1)

**General Comments**

This paper describes the global S and N emissions and deposition in a set of global models run for 2010 under a model intercomparison project (HTAP II) and compares the results with regional monitoring data and previous global modelling. The subject matter is of interest to air quality and ecosystem scientists, particularly those concerned with deposition to oceans, where there is little information from measurements or regional models. I recommend publication with minor revisions, with the most important being an expansion of the dry deposition discussion.

Response: We would like to thank the reviewer for the suggestions to improve this manuscript. Following are the responses to comments.

**Specific comments:**

**Comment:** Section 2.1: It isn't crucial, but it would be helpful to include a brief mention of any other papers (published or forthcoming) that describe additional results from this intercomparison project (e.g. ambient concentrations of particulate matter or O3). With virtual special issues, it's not always obvious where to find related papers.

Response: We added the following paragraph in the manuscript to introduce the related publication from this project.

Line 137-145: Following are some highlight findings in HTAP II. (Stjern et al., 2016) estimated the impact of domestic and foreign emission change of BC, OC and SO4 on regional radiative forcing. (Huang et al., 2017) studied the impact of intercontinental outflow from East Asia to North America on O3 pollution by simulating the regional-scale Sulfur Transport and dEposition Model (STEM) with boundary conditions provided by 3 global transport models. (Jonson et al., 2018) conducted a source apportionment for O3 pollution in Europe and calculated the contributions of emission from global wide. (Tan et al., 2018) investigated the intercontinental export of sulfur and nitrogen emission and its impact on local deposition.

**Comment:** Section 2.2: Can you explain why some values in Tables S1-S3 seem inconsistent? E.g., total S emission from the MMM is 91 Tg in Table S1, but from equation (2) it seems that it should be 55+1+27=83 Tg. Is this an error or am I missing something? Also, the same table seems to show that OsloCTM3 should be excluded for S based on the mass balance criteria described. It's unclear why it is kept. I'm assuming model values that did not meet the criteria are not listed in the tables, though I don't think that was explicitly stated.

Response: Thank you for pointing out this problem. The Multi-model mean of "Emission surface  $SO_2$ " should be 62 instead of 55. Therefore the "total S emission" is 62+1+27=91 Tg. We have changed it in the manuscript.

The tables only list the model values that meet both criteria described in Section 2.2, except the condition that if a model hasn't submitted some important components, which make it impossible to check the criteria. For instance, the 1st criteria compares the global emission with deposition for each model. This criterial is used to check the models that submits the major components of both emission and deposition. The OsloCTM3 model submits the major components of dry and wet deposition. The total value is 40+63=103 Tg. But it hasn't submitted the emission of DMS, and we can't calculate its total S emission. Therefore we can't compare its S deposition with emission. According to the 2nd criteria, we check if the model value is within the range of (median of models  $\pm 1.5 \times$  interquartile) for each component. The components of OsloCTM3 model all pass this quality check. Since the model passes the 2nd criteria and it is unable to check the 1st criteria, we still keep this model.

**Comment:** 1. 265: the 81% value in the text does not match Table 1, which says 61%. Which is correct?

Response: 81% is the average percentage of North America, Europe and Asia. Table 1 gives the separate values for these 3 regions, which are 88%, 75% and 61%, respectively. North America has the 136 stations used for evaluation, more than Europe (82) and East Asia (43), thus the 3 area averaged value is closer to its value.

We have added word in red in the following sentences in the manuscript for clarity: Line243: Overall, 76% of the stations of all networks predicted quantities within  $\pm 50\%$  of observations.

Line 269: Overall, 83% of the MMM results are within  $\pm 50\%$  of observations at stations of all networks.

Line 286: Overall, 81% of the MMM predictions are within  $\pm 50\%$  of observations at stations of all networks.

**Comment**: Il. 267-291 and Table 1: If possible, I suggest adding the number of stations for each comparison to Table 1 since that is likely different as well; where N is relatively low, the number and location of stations used could have a significant impact on the statistics.

Response: We have added the number of stations in Table 1. Because we check the quality and completeness of observation data, the numbers of stations used for evaluation are less than those that are available. We use 136 out of 267 available stations in North America, 82 out of 102 available stations in Europe and 43 out of 52 available stations in Asia.

|                                              |               | North A | America |         |               | Eur    | ope    |         |               | As     | sia    |         |
|----------------------------------------------|---------------|---------|---------|---------|---------------|--------|--------|---------|---------------|--------|--------|---------|
| Wet SO 4 2- Deposition | PhotoCo
mp | HTAP I  | ACCMIP  | HTAP II | PhotoCo
mp | HTAP I | ACCMIP | HTAP II | PhotoCo
mp | HTAP I | ACCMIP | HTAP II |
| Linear Fit Slope                             | 0.9           | 1       | 0.6     | 0.9     | 0.4           | 0.6    | 0.3    | 0.4     | 0.4           | 0.5    | 0.3    | 0.5     |
| Mean Bias                                    | 46.3          | 50      | -18.8   | 30.9    | -67.1         | 51.5   | -125.3 | -31.3   | -218.6        | -182.1 | -292.4 | -161.5  |
| Mean Observation                             | 309.8         | 309.8   | 309.8   | 253.7   | 404.5         | 404.5  | 404.5  | 228.7   | 686.1         | 686.1  | 686.1  | 653.7   |
| Mean Model                                   | 356.1         | 359.8   | 291     | 284.6   | 337.3         | 456.1  | 279.3  | 197.4   | 467.5         | 504.1  | 393.7  | 492.2   |
| R                                            | 0.9           | 0.9     | 0.9     | 0.8     | 0.6           | 0.6    | 0.6    | 0.7     | 0.9           | 0.9    | 0.8    | 0.6     |
| Fraction within ±50%                         | 70.4          | 70      | 72.2    | 76.5    | 78.7          | 52.8   | 78.7   | 86.4    | 80            | 88     | 72     | 68.6    |
| Number of stations                           | 346           | 346     | 346     | 136     | 126           | 126    | 126    | 82      | 49            | 49     | 49     | 43      |
|                                              |               |         |         |         |               |        |        |         |               |        |        |         |
|                                              |               | North A | America |         |               | Eur    | ope    |         |               | As     | sia    |         |
| Wet NO 3 - Deposition  | PhotoCo
mp | HTAP I  | ACCMIP  | HTAP II | PhotoCo
mp | HTAP I | ACCMIP | HTAP II | PhotoCo
mp | HTAP I | ACCMIP | HTAP II |
| Linear Fit Slope                             | 1             | 1       | 0.9     | 1.2     | 0.3           | 0.3    | 0.3    | 0.5     | 0.5           | 0.5    | 0.4    | 0.8     |
| Mean Bias                                    | 34.8          | 21.9    | 44.3    | 57.8    | -41.4         | -60    | -75.2  | -22.0   | -47.8         | -49.3  | -46.4  | -0.8    |
| Mean Observation                             | 191.3         | 191.3   | 191.3   | 153.7   | 300.5         | 300.5  | 300.5  | 237.3   | 263           | 263    | 263    | 356.4   |
| Mean Model                                   | 226.1         | 213.3   | 235.6   | 211.5   | 259.1         | 240.5  | 225.3  | 215.4   | 215.2         | 213.7  | 216.7  | 355.7   |
| R                                            | 0.8           | 0.9     | 0.9     | 0.9     | 0.6           | 0.6    | 0.6    | 0.7     | 0.8           | 0.8    | 0.8    | 0.7     |
| Fraction within ±50%                         | 77            | 84.3    | 68.7    | 66.9    | 75            | 85.2   | 85.2   | 90.2    | 84            | 84     | 88     | 76.7    |
| Number of stations                           | 346           | 346     | 346     | 136     | 126           | 126    | 126    | 82      | 49            | 49     | 49     | 43      |
|                                              |               | North   | mariaa  |         |               | Eur    |        |         |               | ۸.     |        |         |
| Wat NIL + Departies               | DhataCa       | North A | America |         | DhataCa       | Eul    | ope    |         | Dh ata Ca     | As     | sia    |         |
| wei NH 4 Deposition               | mp            | HTAP I  | ACCMIP  | HTAP II | mp            | HTAP I | ACCMIP | HTAP II | mp            | HTAP I | ACCMIP | HTAP II |
| Linear Fit Slope                             | 0.8           | 0.9     | 0.5     | 0.8     | 0.4           | 0.4    | 0.3    | 0.6     | 0.8           | 0.7    | 0.1    | 0.6     |
| Mean Bias                                    | 5.5           | 10.9    | -12.1   | 2.3     | -23.9         | -49.7  | -94.7  | -4.0    | -69.7         | -63.4  | -136.2 | -28.7   |
| Mean Observation                             | 161.3         | 161.3   | 161.3   | 195.5   | 336           | 336    | 336    | 286.1   | 400.5         | 400.5  | 400.5  | 534.5   |
| Mean Model                                   | 166.8         | 172.2   | 149.2   | 197.9   | 312.1         | 286.4  | 241.3  | 282.2   | 330.8         | 337.1  | 264.4  | 505.8   |
| R                                            | 0.9           | 0.9     | 0.8     | 0.9     | 0.8           | 0.6    | 0.6    | 0.6     | 0.8           | 0.8    | 0.2    | 0.7     |
| Fraction within ±50%                         | 82.2          | 84.8    | 75.7    | 87.5    | 73.9          | 79.5   | 78.4   | 75.3    | 76            | 68     | 56     | 60.5    |
| Number of stations                           | 346           | 346     | 346     | 136     | 126           | 126    | 126    | 82      | 49            | 49     | 49     | 43      |

Table 1. Intercomparison of HTAP II MMM performance with previous projects on wet deposition. The unit is mg (N or S) m-2 yr-1.

**Comment:** Section 3.1.2: Additional discussion of dry deposition is warranted, given the large differences with the CASTNET inferential values. The difference between the CASTNET dry deposition calculations and those using the CAPMoN method are touched on (ll. 300-303) but the implications for the models is not fleshed out. I recommend moving this discussion to the end of the section 3.1.2 and discussing the relevance to the ensemble-measurement comparison. How does the CASTNET dry deposition velocity parametrization compare with those used in the various models? How do the modelled air concentrations of SO2, HNO3, etc. compare with the CASTNET observations?

Response: Thank you for this useful suggestion. We have moved the discussion of uncertainty of CASNET to the end of section 3.1.2 as an explanation of the model bias. We also compare the air concentrations and dry deposition velocities between the models and CASTNET dataset in the manuscript as follows:

Line 349-381: Since the CASTNET dry deposition is not actually measured data but instead a combination of measured concentration of species and modelled dry deposition velocities, it is necessary to investigate which factor of these two contributes to the model bias. We compare the modelled air pollutant concentrations with CASENET measurements as shown in Table S4-S8. The MMM overestimates the SO2, SO42-, HNO3, NO3- and NH4+ concentrations by 394%, 40%, 217%, 135% and 173%, respectively. It should be noted that the CASTNET sites are generally located in rural regions that are away from emission sources (Sickles and Shadwick, 2008), thus the measured concentrations of air pollutants are relatively low compared with those of urban sites. While the resolutions of the HTAP II models range from 0.5° to 3°, and are not fine enough to reproduce the characteristic of some rural sites. The models with finer resolutions except CHASER\_t106 model (i.e. EMEP\_rv48 (0.5 × 0.5) and SPRINTARS (1.1 × 1.1)) generally perform better than the others, while models with coarse resolutions (i.e. CHASER\_re1 (2.8 × 2.8) and OsloCTM3.v2) are generally not performing well for all species. This could explain the overestimation of air pollutant concentrations at the CASTNET sites.

In order to check the differences of modelled dry deposition velocity between CASNET and HTAP II models, we adopt the general approach for calculating dry deposition velocity from (Wesely, 1989).

$$V_{\rm d} = - F_{\rm c} / C_{\rm a} \tag{7}$$

Where  $V_d$  is the deposition velocity,  $F_c$  is the dry deposition flux and  $C_a$  is the concentration of species. The negative mark indicates the direction of the dry deposition velocity. This scheme has been widely adopted in global models (Wesely and Hicks, 2000) with modifications. We compare the calculated dry deposition velocity of models and CASTNET (Table S9-S13). The mean bias of dry deposition velocities for MMM are -8%, 0.3%, 7%, 19% and 2% for SO2, SO42-, HNO3, NO3- and NH4+, respectively, which are much lower than those of air pollutants. The model bias for dry deposition at the CASTNET sites mainly comes from the model over prediction of air pollutant concentration.

Table S4. Multi-model performance on simulating SO2 concentration at CASTNET sites. The unit is  $\mu g$  (S) m-3.

| Species          | CAM-chem | CHASER_r
e1 | CHASER_t106 | EMEP_rv48 | GEOSCHEMAD
JOINT | GOCART | OsloCTM3
.v2 | SPRINTARS | MMM  |
|------------------|----------|----------------|-------------|-----------|---------------------|--------|-----------------|-----------|------|
| Mean Observation | 0.82     | 0.82           | 0.82        | 0.82      | 0.82                | 0.82   | 0.82            | 0.82      | 0.82 |
| Mean Model       | 5.31     | 5.79           | 5.51        | 1.71      | 5.36                | 2.49   | 4.61            | 1.72      | 4.06 |
| Linear Fit Slope | 6.77     | 6.65           | 7.90        | 2.22      | 6.33                | 2.90   | 5.43            | 2.03      | 5.03 |

| Mean Bias          | 0.00                        | 0.00                             | 0.00                             | 0.00                        | 0.00                             | 0.00                             | 0.00                        | 0.00                             | 0.00   |
|--------------------|-----------------------------|----------------------------------|----------------------------------|-----------------------------|----------------------------------|----------------------------------|-----------------------------|----------------------------------|--------|
| Bias% 1 | 546.66                      | 604.77                           | 570.95                           | 107.92                      | 553.04                           | 203.68                           | 460.86                      | 109.94                           | 394.73 |
| R                  | 0.79                        | 0.81                             | 0.84                             | 0.89                        | 0.76                             | 0.88                             | 0.84                        | 0.76                             | 0.90   |
| F                  | 12.50                       | 6.25                             | 12.50                            | 31.25                       | 16.25                            | 21.25                            | 15.00                       | 43.75                            | 11.25  |
| NMB                | 546.66                      | 604.77                           | 570.95                           | 107.92                      | 553.04                           | 203.68                           | 460.86                      | 109.94                           | 394.73 |
| NME                | 548.91                      | 606.20                           | 573.30                           | 116.37                      | 554.09                           | 208.21                           | 462.33                      | 117.67                           | 396.47 |
| MFB                | 104.46                      | 119.52                           | 101.46                           | 21.00                       | 110.67                           | 65.28                            | 105.73                      | 25.09                            | 99.26  |
| MFE                | 116.11                      | 125.36                           | 110.97                           | 69.95                       | 113.81                           | 88.82                            | 111.67                      | 61.69                            | 106.31 |
| Number of stations | 80                          | 80                               | 80                               | 80                          | 80                               | 80                               | 80                          | 80                               | 80     |
| Spatial resolution | $1.9^\circ 	imes 2.5^\circ$ | $2.8^{\circ} \times 2.8^{\circ}$ | $1.1^{\circ} \times 1.1^{\circ}$ | $0.5^\circ 	imes 0.5^\circ$ | $2.0^{\circ} \times 2.5^{\circ}$ | $1.3^{\circ} \times 1.0^{\circ}$ | $2.8^\circ 	imes 2.8^\circ$ | $1.1^{\circ} \times 1.1^{\circ}$ |        |

1 Bias is calculated by dividing Mean Bias with Mean Observation. The unit is %.

| Species            | CAM-chem                    | CHASER_re1                  | CHASER_t106                      | EMEP_rv48                    | GEOSCHEMADJOINT                  | OsloCTM3.v2                 | MMM   |
|--------------------|-----------------------------|-----------------------------|----------------------------------|------------------------------|----------------------------------|-----------------------------|-------|
| Mean Observation   | 0.64                        | 0.64                        | 0.64                             | 0.64                         | 0.64                             | 0.64                        | 0.64  |
| Mean Model         | 1.06                        | 1.24                        | 1.09                             | 0.74                         | 0.70                             | 0.52                        | 0.89  |
| Linear Fit Slope   | 1.97                        | 1.90                        | 1.80                             | 1.62                         | 1.17                             | 0.96                        | 1.57  |
| Mean Bias          | 0.00                        | 0.00                        | 0.00                             | 0.00                         | 0.00                             | 0.00                        | 0.00  |
| Bias% 1 | 66.99                       | 94.61                       | 71.61                            | 16.30                        | 10.81                            | -18.18                      | 40.36 |
| R                  | 0.93                        | 0.84                        | 0.88                             | 0.92                         | 0.92                             | 0.87                        | 0.91  |
| F                  | 41.25                       | 26.25                       | 40.00                            | 57.50                        | 92.50                            | 82.50                       | 63.75 |
| NMB                | 66.99                       | 94.61                       | 71.61                            | 16.30                        | 10.81                            | -18.18                      | 40.36 |
| NME                | 72.20                       | 95.50                       | 73.78                            | 40.59                        | 23.01                            | 25.30                       | 46.73 |
| MFB                | 34.17                       | 59.16                       | 45.63                            | -18.17                       | 6.00                             | -32.63                      | 23.84 |
| MFE                | 46.67                       | 60.54                       | 49.32                            | 50.23                        | 23.21                            | 37.75                       | 33.98 |
| Number of stations | 80                          | 80                          | 80                               | 80                           | 80                               | 80                          | 80    |
| Spatial resolution | $1.9^\circ 	imes 2.5^\circ$ | $2.8^\circ 	imes 2.8^\circ$ | $1.1^{\circ} \times 1.1^{\circ}$ | $0.5^\circ \times 0.5^\circ$ | $2.0^{\circ} \times 2.5^{\circ}$ | $2.8^\circ 	imes 2.8^\circ$ |       |

Table S5. Same as Table S4 but for  ${\rm SO_4}^{2\text{-}}$  concentration. The unit is  $\mu g$  (S)  $m^{\text{-3}}.$

Table S6. Same as Table S4 but for  $HNO_3$  concentration. The unit is  $\mu g \left( N \right) m^{\text{-3}}.$

| Species            | CAM-chem                    | CHASER_re1                  | CHASER_t106                      | EMEP_rv48                    | GEOSCHEMADJOINT                  | OsloCTM3.v2                 | MMM    |
|--------------------|-----------------------------|-----------------------------|----------------------------------|------------------------------|----------------------------------|-----------------------------|--------|
| Mean Observation   | 0.17                        | 0.17                        | 0.17                             | 0.17                         | 0.17                             | 0.17                        | 0.17   |
| Mean Model         | 0.64                        | 0.83                        | 0.71                             | 0.36                         | 0.64                             | 0.14                        | 0.55   |
| Linear Fit Slope   | 3.34                        | 4.40                        | 5.10                             | 2.00                         | 2.48                             | 0.72                        | 3.01   |
| Mean Bias          | 0.00                        | 0.00                        | 0.00                             | 0.00                         | 0.00                             | 0.00                        | 0.00   |
| Bias% 1 | 264.73                      | 376.04                      | 309.11                           | 106.39                       | 264.54                           | -19.82                      | 216.83 |
| R                  | 0.78                        | 0.65                        | 0.82                             | 0.85                         | 0.74                             | 0.76                        | 0.84   |
| F                  | 3.75                        | 2.50                        | 1.25                             | 28.75                        | 1.25                             | 76.25                       | 3.75   |
| NMB                | 264.73                      | 376.04                      | 309.11                           | 106.39                       | 264.54                           | -19.82                      | 216.83 |
| NME                | 265.39                      | 376.18                      | 309.11                           | 106.84                       | 264.54                           | 30.38                       | 216.83 |
| MFB                | 107.09                      | 117.52                      | 105.97                           | 60.13                        | 113.19                           | -27.44                      | 98.66  |
| MFE                | 107.50                      | 117.59                      | 105.97                           | 61.55                        | 113.19                           | 42.18                       | 98.66  |
| Number of stations | 80                          | 80                          | 80                               | 80                           | 80                               | 80                          | 80     |
| Spatial resolution | $1.9^\circ 	imes 2.5^\circ$ | $2.8^\circ 	imes 2.8^\circ$ | $1.1^{\circ} \times 1.1^{\circ}$ | $0.5^\circ \times 0.5^\circ$ | $2.0^{\circ} \times 2.5^{\circ}$ | $2.8^\circ 	imes 2.8^\circ$ |        |

Table S7. Same as Table S4 but for  $\mathrm{NO_3}^-$  concentration. The unit is  $\mu g$  (N)  $m^{\text{-3}}.$

| Species            | EMEP_rv48 | GEOSCHEMADJOINT | MMM    |
|--------------------|-----------|-----------------|--------|
| Mean Observation   | 0.17      | 0.17            | 0.17   |
| Mean Model         | 0.17      | 0.63            | 0.40   |
| Linear Fit Slope   | 0.67      | 2.58            | 1.63   |
| Mean Bias          | 0.00      | 0.00            | 0.00   |
| Bias% 1 | 0.05      | 270.05          | 135.05 |
| R                  | 0.80      | 0.74            | 0.77   |
| F                  | 65.00     | 12.50           | 17.50  |

| NMB                | 0.05                             | 270.05                           | 135.05 |
|--------------------|----------------------------------|----------------------------------|--------|
| NME                | 35.60                            | 279.48                           | 144.21 |
| MFB                | 2.19                             | 103.34                           | 76.92  |
| MFE                | 41.98                            | 113.85                           | 87.53  |
| Number of stations | 80.0                             | 80                               | 80     |
| Spatial resolution | $0.5^{\circ} \times 0.5^{\circ}$ | $2.0^{\circ} \times 2.5^{\circ}$ |        |

Table S8. Same as Table 4 but for  $NH_4^+$  concentration. The unit is  $\mu g$  (N) m-3.

| Species            | EMEP_rv48                        | GEOSCHEMADJOINT                  | MMM    |
|--------------------|----------------------------------|----------------------------------|--------|
| Mean Observation   | 0.56                             | 0.56                             | 0.56   |
| Mean Model         | 1.13                             | 1.94                             | 1.54   |
| Linear Fit Slope   | 2.00                             | 3.47                             | 2.74   |
| Mean Bias          | 0.00                             | 0.00                             | 0.00   |
| Bias% 1 | 101.13                           | 244.46                           | 172.79 |
| R                  | 0.91                             | 0.94                             | 0.95   |
| F                  | 26.25                            | 1.25                             | 5.00   |
| NMB                | 101.13                           | 244.46                           | 172.79 |
| NME                | 101.89                           | 244.46                           | 172.79 |
| MFB                | 58.09                            | 106.87                           | 88.42  |
| MFE                | 59.91                            | 106.87                           | 88.42  |
| Number of stations | 80                               | 80                               | 80     |
| Spatial resolution | $0.5^{\circ} \times 0.5^{\circ}$ | $2.0^{\circ} \times 2.5^{\circ}$ |        |

Table S9. Comparison of dry deposition velocity of SO2 between models and CASTNET. The unit is cm s-1.

|                   | CAM-chem | CHASER_r
e1 | CHASER_t
106 | EMEP_rv4 | GEOSCHE
MADJOIN
T | GOCART | OsloCTM3.
v2 | SPRINTAR
S | MMM   |
|-------------------|----------|----------------|-----------------|----------|-------------------------|--------|-----------------|---------------|-------|
| mean obs          | 0.27     | 0.27           | 0.27            | 0.27     | 0.27                    | 0.27   | 0.27            | 0.27          | 0.27  |
| mean model        | 0.14     | 0.17           | 0.16            | 0.36     | 0.33                    | 0.41   | 0.39            | 0.50          | 0.24  |
| Linear Fit Slope  | 0.16     | 0.02           | -0.01           | -0.35    | 0.20                    | 0.03   | -0.62           | 0.15          | 0.04  |
| mean bias         | -0.13    | -0.09          | -0.11           | 0.10     | 0.07                    | 0.15   | 0.12            | 0.23          | -0.02 |
| bias%             | -47.79   | -34.88         | -41.44          | 35.54    | 25.05                   | 54.32  | 46.08           | 87.86         | -8.39 |
| R                 | 0.18     | 0.03           | -0.02           | -0.23    | 0.09                    | 0.03   | -0.44           | 0.23          | 0.06  |
| F                 | 36.25    | 60.00          | 53.75           | 56.25    | 70.00                   | 42.50  | 50.00           | 30.00         | 77.50 |
| NMB               | -47.79   | -34.88         | -41.44          | 35.54    | 25.05                   | 54.32  | 46.08           | 87.86         | -9.03 |
| NME               | 56.70    | 44.51          | 49.62           | 62.93    | 55.54                   | 60.74  | 68.83           | 88.07         | 34.88 |
| MFB               | -67.47   | -39.46         | -48.79          | 27.96    | 16.61                   | 45.18  | 36.33           | 65.08         | -3.83 |
| MFE               | 75.10    | 52.75          | 60.26           | 50.44    | 42.03                   | 49.40  | 53.58           | 65.21         | 35.00 |
| Number of station | 80.00    | 80.00          | 80.00           | 80.00    | 80.00                   | 80.00  | 80.00           | 80.00         | 80    |

Table S10. Same as Table S9 but for dry deposition velocity of  $SO_4^{2-}$

|                  | 1001     |            | 14010 57 0401 | or ary acpos |                 |             |       |
|------------------|----------|------------|---------------|--------------|-----------------|-------------|-------|
|                  | CAM-chem | CHASER_re1 | CHASER_t106   | EMEP_rv48    | GEOSCHEMADJOINT | OsloCTM3.v2 | MMM   |
| mean obs         | 0.13     | 0.13       | 0.13          | 0.13         | 0.13            | 0.13        | 0.13  |
| mean model       | 0.15     | 0.09       | 0.10          | 0.17         | 0.13            | 0.21        | 0.13  |
| Linear Fit Slope | 0.29     | 0.01       | -0.01         | 0.30         | 0.11            | 0.21        | 0.10  |
| mean bias        | 0.02     | -0.03      | -0.03         | 0.04         | 0.01            | 0.08        | 0.00  |
| Bias %           | 16.15    | -26.92     | -24.39        | 31.34        | 4.06            | 63.18       | 0.30  |
| R                | 0.42     | 0.07       | -0.16         | 0.17         | 0.28            | 0.27        | 0.34  |
| F                | 83.75    | 82.50      | 85.00         | 67.50        | 90.00           | 36.25       | 88.75 |

| NMB                | 16.15 | -26.92 | -24.39 | 31.34 | 4.06  | 63.18 | 0.30  |
|--------------------|-------|--------|--------|-------|-------|-------|-------|
| NME                | 26.38 | 32.21  | 30.95  | 45.91 | 22.36 | 63.67 | 21.12 |
| MFB                | 17.60 | -26.05 | -22.58 | 23.00 | 7.78  | 50.30 | 4.56  |
| MFE                | 26.19 | 34.91  | 33.14  | 37.74 | 22.75 | 50.56 | 21.73 |
| Number of stations | 80    | 80     | 80     | 80    | 80    | 80    | 80    |

**Table S11. Same as Table S9 but for dry deposition velocity of HNO3**

|                    | CAM-chem | CHASER_re1 | CHASER_t10
6 | EMEP_rv48 | GEOSCHEM
ADJOINT | OsloCTM3.v2 | MMM   |
|--------------------|----------|------------|-----------------|-----------|---------------------|-------------|-------|
| mean obs           | 1.34     | 1.34       | 1.34            | 1.34      | 1.34                | 1.34        | 1.34  |
| mean model         | 0.68     | 1.07       | 1.41            | 1.35      | 1.51                | 3.55        | 1.25  |
| Linear Fit Slope   | 0.03     | 0.06       | -0.37           | -0.03     | -0.40               | -1.24       | -0.10 |
| mean bias          | -0.66    | -0.28      | 0.06            | 0.01      | 0.17                | 2.21        | -0.10 |
| Bias %             | -49.12   | -20.70     | 4.60            | 0.74      | 12.44               | 164.52      | -7.27 |
| R                  | 0.10     | 0.05       | -0.28           | -0.03     | -0.29               | -0.38       | -0.12 |
| F                  | 50.00    | 72.50      | 77.50           | 85.00     | 76.25               | 16.25       | 85.00 |
| NMB                | -49.12   | -20.70     | 4.60            | 0.74      | 12.44               | 164.52      | -7.27 |
| NME                | 49.92    | 34.65      | 36.69           | 29.96     | 38.75               | 165.99      | 27.47 |
| MFB                | -62.09   | -27.68     | 2.14            | -0.16     | 9.65                | 84.58       | -7.01 |
| MFE                | 63.78    | 42.88      | 37.23           | 30.98     | 36.88               | 85.84       | 29.52 |
| Number of stations | 80       | 80         | 80              | 80        | 80                  | 80          | 80    |

**Table S12. Same as Table S9 but for dry deposition velocity of NO3-**

|                    | EMEP_rv48 | GEOSCHEMADJOINT | MMM   |
|--------------------|-----------|-----------------|-------|
| mean obs           | 0.12      | 0.12            | 0.12  |
| mean model         | 0.29      | 0.10            | 0.14  |
| Linear Fit Slope   | 0.65      | 0.00            | 0.06  |
| mean bias          | 0.17      | -0.02           | 0.02  |
| Bias %             | 146.99    | -16.50          | 18.69 |
| R                  | 0.26      | 0.00            | 0.05  |
| F                  | 7.50      | 77.50           | 73.75 |
| NMB                | 146.99    | -16.50          | 18.69 |
| NME                | 147.73    | 40.11           | 37.35 |
| MFB                | 81.93     | -17.07          | 17.29 |
| MFE                | 82.58     | 38.99           | 32.05 |
| Number of stations | 80        | 80              | 80    |

**Table S13. Same as Table S9 but for dry deposition velocity of $NH_4^+$**

|                    | CAM-chem | GEOSCHEMADJOINT | MMM   |
|--------------------|----------|-----------------|-------|
| mean obs           | 0.12     | 0.12            | 0.12  |
| mean model         | 0.22     | 0.06            | 0.12  |
| Linear Fit Slope   | 0.38     | 0.06            | 0.13  |
| mean bias          | 0.10     | -0.06           | 0.00  |
| Bias               | 81.91    | -47.39          | -1.72 |
| R                  | 0.40     | 0.23            | 0.27  |
| F                  | 15.00    | 60.00           | 87.50 |
| NMB                | 81.91    | -47.39          | -1.72 |
| NME                | 82.81    | 48.76           | 22.13 |
| MFB                | 60.11    | -57.09          | 2.01  |
| MFE                | 60.62    | 59.99           | 22.67 |
| Number of stations | 80       | 80              | 80    |

**Comment:** 1. 347: Australia appears to receive higher coastal S deposition than E. Asia, so should be listed here as well.

Response: Coastal Australia emitted 2.0 Tg(S) yr-1 of S emission, higher than that of coastal East Asia (1.8 Tg(S) yr-1). But coastal Australia received 1.5 Tg(S) yr-1 of S deposition, lower than that of coastal East Asia (2.9 Tg(S) yr-1). This is because the high S emission emitted in non-coastal East Asia (15.0 Tg(S) yr-1) brings deposition to its coastal region via long-range transport. While lower S emission in non-coastal Australia (1.5 Tg(S) yr-1) has less impact on its coastal region.

**Comment**: 1. 374: Why the 32% increase in ocean S emissions? Is that real or the result of improved emission budgets?

Response: The total S emission in HTAP I is 91 Tg(S) in 2001, of which 66.4 Tg(S) is  $SO_2$  emission, 6.3 Tg(S) is  $SO_4^{2-}$  emission and 18.2 Tg(S) is DMS emission. The total S emission in HTAP II is 91 Tg(S) in 2010, of which 63 Tg(S) is  $SO_2$  emission, 1 Tg(S) is  $SO_4^{2-}$  emission and 27 Tg(S) is DMS emission.

The amount of total S emission and SO2 emission are similar between HTAP I and HTAP II. While the  $SO_4^{2^2}$  emission is decreased by 5 Tg(S) and DMS emission is increased by 9 Tg(S). Since the DMS emission is generally from coastal and ocean sources, the large difference of oceanic S emissions comes from the DMS emission.

We compare the emission of DMS with literatures. The range of DMS is estimated to be 23-35 Tg(S) by (Simo and Dachs, 2002) from remote sensing of biogeophysical data and to be about 28 Tg(S) estimated by (Kloster et al., 2006). The 27 Tg(s) of HTAP II is closer to the abovementioned range, and the 18 Tg(s) of HTAP I could be slightly underestimated.

Another possible reason is the calculation of multi-model mean of DMS emission. The following table listed the S emission by different models. Although all models except EMEP\_rv48 and GEMMACH are confirmed to include DMS emission in simulations, but only 5 out of 10 models have submitted the DMS emission. The relative low number of submission could cause uncertainty in calculating the multi-model ensemble of DMS emission.

| Model/Species    | DMS       |
|------------------|-----------|
| wodel/species    | DIVIS     |
| CAMChem          | 28        |
| CHASER_re1       | 25        |
| CHASER_t106      | 23        |
| EMEP_rv48        | Not used  |
| GEMMACH          | Not given |
| GEOS5            | 31        |
| GEOSCHEMADJOINT  | Include   |
| OsloCTM3.v2      | Include   |
| GOCARTv5         | Include   |
| SPRINTARS        | 22        |
| C-IFS_v2         | Include   |
| Multimodel mean* | 27        |
|                  |           |

Table 1. Summary of Global Emission of S in 2010 (Tg(S) yr-1

**Comment:** Il. 520-536: There is discussion of the areas of increasing NHx ratio, but globally there appears to be a general decrease (e.g. over the oceans). Maybe add a comment on this.

Response: Thank you for your suggestion. We add the following sentence in the manuscript.

Line 570-572: Generally, we found a 10% decrease in the ratio of NHx deposition from 2001 to 2010. In particular, a 30% decrease in the ratio of NHx is found in southeastern China, mainly due to the large increase in NOx emission during the last decade.

**Comment:** Fig. 2: Observation (point) values are very difficult to see on these small plots. Can they be enlarged, since the discussion in 3.1.1 hinges on the regional comparison? Fig. 4 is better; I would suggest that size is the minimum needed.